# Design, Synthesis and Biological Evaluation of 6-(Imidazo[1,2-a]pyridin-6-yl)quinazoline Derivatives as Anticancer Agents via PI3Kα Inhibition

**DOI:** 10.3390/ijms24076851

**Published:** 2023-04-06

**Authors:** Mei Li, Daoping Wang, Qing Li, Fang Luo, Ting Zhong, Hongshan Wu, Liang Xiong, Meitao Yuan, Mingzhi Su, Yanhua Fan

**Affiliations:** 1State Key Laboratory for Functions and Applications of Medicinal Plants, Guizhou Medical University, Guiyang 550014, China; 2The Key Laboratory of Chemistry for Natural Products of Guizhou Province and Chinese Academy of Sciences, Guiyang 550014, China

**Keywords:** cell cycle arrest, cell apoptosis, PI3Kα inhibitor, quinazoline, imidazo[1,2-a]pyridine

## Abstract

Aberrant expression of the phosphatidylinositol 3-kinase (PI3K) signalling pathway is often associated with tumourigenesis, progression and poor prognosis. Hence, PI3K inhibitors have attracted significant interest for the treatment of cancer. In this study, a series of new 6-(imidazo[1,2-a]pyridin-6-yl)quinazoline derivatives were designed, synthesized and characterized by ^1^H NMR, ^13^C NMR and HRMS spectra analyses. In the in vitro anticancer assay, most of the synthetic compounds showed submicromolar inhibitory activity against various tumour cell lines, among which **13k** is the most potent compound with IC_50_ values ranging from 0.09 μΜ to 0.43 μΜ against all the tested cell lines. Moreover, **13k** induced cell cycle arrest at G2/M phase and cell apoptosis of HCC827 cells by inhibition of PI3Kα with an IC_50_ value of 1.94 nM. These results suggested that compound **13k** might serve as a lead compound for the development of PI3Kα inhibitor.

## 1. Introduction

Phosphatidylinositol 3-kinase (PI3K) is a lipid kinase that plays a key regulatory role in various cellular physiological processes including cell growth, proliferation, survival and metabolism [1,2]. Akt (protein kinase B, PKB) is a serine/threonine kinase and participates in the key role of the PI3K signalling pathway. Research shows that mutations and abnormal activation of the PI3K-AKT pathway are often identified as one of the major factors resulted in tumourigenesis, progression and poor prognosis [3,4,5]. PI3K is usually divided into three categories (classes I, II and III) [6]. PI3Kα belongs to class I, which mainly consists of a regulatory subunit (p85) and a catalytic subunit (p110) [7]. The mutation of PIK3CA, the encoding gene of PI3Kα, is one of the most common mutations in tumours and would result in the under-expression or absence of PTEN (phosphatase and tensin homolog) and hyperactivation of PI3K downstream signalling pathways [8,9]. Due to the critical roles of PI3K signalling pathway in tumour occurrence, development and drug resistance, inhibitors targeting PI3K have attracted widespread attention [10,11]. Currently, dozens of subtype-selective and pan-PI3K inhibitors are in various stages of clinical studies for the treatment of human malignancies, yet the discovery of additional lead compounds for novel PI3Kα inhibitors with better efficacy and less toxic side effects remains an urgent therapeutic need [12,13,14].

Quinazolines are the major compounds in the aromatic backbone of nitrogen-containing heterocyclic compounds with a wide range of biological activities such as anti-inflammatory, antimicrobial, antimalarial and antitumour [15,16,17]. In particular, many drugs containing 4-aminoquinazoline structures have been reported to exhibit prominent antitumour activity through various mechanisms [18,19,20,21]. In recent years, it has been shown that 4-aminoquinazoline derivatives show good antitumour activity by inhibiting PI3Kα [22]. This shows that 4-aminoquinazolines are an important class of molecular scaffold that can be used for the development of antitumour drugs.

In a previous study, we designed and synthesised a series of 4-aminoquinazoline derivatives and obtained a compound **6b** as a PI3Kα inhibitor [23]. Based on the previous structure activity relationships (SAR) analysis and pharmacophore fusion strategy, structure modification of **6b** was performed to further improve the activity. According to the SAR analysis, 4-aminoquinazoline derivative moiety is the main critical pharmacophore of **6b** for its PI3Kα inhibitory activity. Therefore, this moiety was retained as the basic scaffold for our target compound. Since imidazo[1,2-a]pyridine, the key pharmacodynamic group of PI3Kα inhibitors TAK-117 and HS-173, is an important class of nitrogen-containing fused heterocyclics compounds that can effectively inhibit the growth of cancer cells, it was introduced to the position 6 of 4-aminoquinazoline [24,25,26,27,28]. Herein, a series of 6-(imidazo[1,2-a]pyridin-6-yl)quinazoline derivatives were designed and synthesized (Figure 1), and biological evaluation was performed to verify their PI3Kα inhibitory activities and antitumour effects.

## 2. Results and Discussion

### 2.1. Chemistry

The synthetic route for intermediates **7a**–**o** of target products **10a**–**u** is shown in Figure 1. A purchased raw material, 6-iodoquinazoline 4-3(H)-one was chlorinated in POCl_3_ in the presence of DIPEA to give intermediate **2**. Intermediates **5a**–**q** were obtained by nucleophilic substitution reaction with primary or secondary amines, which subsequently reacted with 2-aminopyridine-5-boronic pinacol ester acid by Suzuki–Miyaura cross-coupling reaction to give intermediates **7a**–**o**. Intermediates **7a**–**o** were cyclized with methyl bromopyruvate or ethyl bromopyruvate to give the target products **10a**–**u**, as shown in Figure 2. To improve the inhibitory activities of the target compounds, we performed further optimization of the substituents. Unfortunately, when the ester side chain was replaced with a cycloalkane, we failed to yield our target products by Figure 2, so we opted for an alternative synthetic route. As shown in Figure 3, intermediate **6** reacted with compound **11** to afford compound **12**, which was coupled with intermediates **5** to give our target products **13a**–**k** by Suzuki–Miyaura cross-coupling reaction. In this thesis, we introduced different substituents at the C^6^ and C^4^ positions of the 4-aminoquinazoline backbone and synthesised various ester and amines to further explore their possible structure–activity relationship (SAR), and all compounds are shown in Table 1.

### 2.2. Biological Evaluation

#### 2.2.1. Antiproliferation Activity Assay

To test the antiproliferative activity of all target compounds, IC_50_ values were measured by MTT assay on various cancer cell lines including HCC827 (human non-small cell lung cancer cells), A549 (human non-small cell lung cancer cells), SH-SY5Y (human neuroblastoma cells), HEL (human erythroid and leukocyte leukaemia cells) and MCF-7 (human breast cancer cells). As shown in Table 1, most of the compounds showed significant antiproliferative activity in all the test cancer cells. Notably, most of the active compounds were more sensitive to HCC827 cells. In addition to HCC827 cells, PI3K was also overexpressed in other tested cells [29,30,31,32]. As to the reasons for the different sensitivity of the compounds to these tested cells, we hypothesized it might be because the PI3K pathway is not as equally important in the survival and proliferation of these cells as it is in HCC827 cells. For example, when PI3K signalling pathway is inhibited in A549 cells, cells can still maintain cell survival and proliferation through Ras/MERK/ERK pathway [33], which hence leads to different inhibitory activities of PI3K inhibitors in these two cells. According to the data of the antiproliferative assay, we conclude the following structure activity relationship. (I) In general, the antiproliferative activity of the compounds significantly decreased when R_1_ substituent group was an alkyl, suggesting that simultaneous alkylation of NH_2_ at the 4-position of quinazoline would impair the antiproliferative activity of the target compounds. (II) When R_3_ = COOCH_3_, most of the compounds are more active than R_3_ = COOC_2_H_5_, such as compounds **10q** and **10h**, **10r** and **10i**, and when R_3_ = COOC_2_H_5_ and R_2_ is pyridine, the ortho-nitrogen is more active than meta-nitrogen. (III) The activity of the compounds was generally increased when benzene was introduced into the R_3_, as in **13c** and **10l**, **13a** and **10r**, but a decrease in activity was found with the introduction of the electron withdrawing group F on the R_3_-substituted benzene, as in compounds **13a** and **13b**, **13c** and **13d**. Overall, compound **13k** showed the best antiproliferative activity against HCC827 cells with an IC_50_ value of 0.09 μM, which could be attributed to the conventional hydrogen bond formed between the R_2_-substituted tetrahydropyran and residue Gln859 in the active site of the target proprotein. To evaluate the selectivity of **13k** on cancer cells, the cytotoxicity of **13k** on human normal cell MRC-5 (human embryonic lung fibroblasts) was determined. Compound **13k** showed much less antiproliferative activity against MRC-5 with an IC_50_ value of 1.98 μM, which is more than 20-fold different from HCC827 cells (Table 2). Moreover, as shown in Figure 2, **13k** treatment time-dependently inhibited the proliferation of HCC827 cells. Taken together, we chose HCC827 cells to further explore the anticancer effects and mechanisms of **13k**.

#### 2.2.2. Compound **13k** Inhibits PI3Kα and Blocks the PI3K Pathway in HCC827 Cells

To evaluate the in vitro kinase inhibitory activity of **13k** against PI3Kα, the kinase activity of PI3Kα was tested using the ADP-Glo^TM^ Max Assay method. HS-173, a known PI3Kα inhibitor, was used as a positive control. As shown in Table 3, **13k** significantly inhibited the kinase activity of PI3Kα with an IC_50_ value of 1.94 nM. This suggests that compound **13k** is a potential PI3Kα inhibitor.

Aberrant expression of PI3K signalling pathway is closely related to the process of tumourigenesis [34]. Lung cancer is the most lethal malignancy in the world, with non-small cell lung cancer (NSCLC) being the most commonly reported histological subtype [35]. According to reports, new oncogene changes have been discovered in NSCLC, including genetic changes in the PI3K pathway, and PIK3CA mutations in NSCLC may co-occur with epidermal growth factor receptor (EGFR), Kirsten rat sarcoma viral oncogene homologue (KRAS) and anaplastic lymphoma kinase (ALK) mutations [36,37]. Therefore, we chose compound **13k** to investigate the mechanism of this compound in HCC827 cells. Since **13k** significantly inhibited PI3Kα activity, we further verified the effects of **13k** on the PI3K/AKT pathway by Western blot. As shown in Figure 3, the phosphorylation level of PI3K was significantly reduced after **13k** treatment in a dose-dependent manner. The phosphorylation levels of its downstream proteins, AKT, mTOR and GSK3β, were correspondingly reduced. The results confirmed the inhibitory effect of **13k** on PI3K pathway. The AKT/MAPK signalling pathway, downstream of PI3K, is considered a classical cancer signalling pathway and is involved in the development of many cancers [38,39,40]. Hence, PI3K inhibitors usually also affect the activation of three major categories of MAPK including ERK, JNK and p38 [41]. As shown in Figure 4, the p-JNK/JNK and p-p38/p38 values of HCC827 cells after **13k** treatment were significantly higher than those of the control group, indicating that **13k** can regulate the MAPK pathway through AKT.

#### 2.2.3. Molecular Docking Study of Compound **13k**

Molecular docking simulations were performed to investigate the binding mode between **13k** and its target protein PI3Kα (PDB code: 4ZOP). Similar to the binding mode of PI3Kα inhibitor previously discovered, **13k** formed two conventional hydrogen bonds with the residues Lys802 and Gln859 as well as hydrophobic interactions including van der Waals, pi–pi T-shaped and pi–sulfur interactions in the active site of PI3Kα. As shown in Figure 5, the benzene ring of compound **13k** also formed a pi–alkyl interaction with Leu807 disability. The results indicated that **13k** could engage the ATP-binding pocket of PI3Kα. In addition, **13k** also formed similar hydrophobic interactions with residues in the acetyl–lysine binding sites.

#### 2.2.4. Compound **13k** Induced G2/M Phase Block in HCC827 Cells

It has shown that the anti-proliferative activity of PI3Kα inhibitors was associated with cell cycle arrest [42]. Therefore, we examined the effects of **13k** on cell cycle distribution. As shown in Figure 6, **13k** treatment for 48 h resulted in a significant G2/M phase block of HCC827 cells (52.21%), when compared to the control group (20.84%). In order to elucidate the potential regulatory mechanism of **13k** on cell cycle, proteins associated with cell cycle regulation were detected using Western blot. As described in Figure 6C–G, the protein levels of cyclin B1, c-Myc and CDK1 were dose-dependently decreased by **13k** treatment. Additionally, both the total and phosphorylated proteins of CHK1 and CDC25A were also reduced by compound **13k**.

#### 2.2.5. Compound 13k Induced Cell Apoptosis

To further investigate the effects of **13k** on apoptosis, cells were treated with various doses of **13k** ranging from 0 to 0.32 μM. The percentage of apoptotic HCC827 cells was detected using Annexin V-FITC /PI double staining. The results showed that **13k** dose-dependently induced cellular apoptosis from 1.73–37.61%. In addition, Hoechst 33342 staining analysis indicated **13k** treatment caused cell shrinkage and DNA fragmentation, which resulted in an enhanced absorption and intensity of Hoechst staining. To further elucidate the mechanism of **13k**-induced apoptosis, the apoptosis-related protein levels was examined by Western blot. We found that compound **13k** increased the protein levels of cleaved caspase-9 and cleaved PARP in a concentration-dependent manner, while the ratios of Bax/Bcl-2 were upregulated, further indicating that compound **13k** promotes cell apoptosis (Figure 7).

#### 2.2.6. In 3D Spheroid Cell Inhibition Assay

The 3D cell culture has been proved to more realistically reproduce the interactions between cell–cell and cell–extracellular matrix interactions and more accurately simulate the actual microenvironment of cells in tissues [43,44,45]. These allow the cell behaviour characteristics of cells in 3D cell culture to be closer to the survival state in living organisms. Hence, it was widely applied in research fields including new drug screening, tumour cell system biology, stem cell research and functional tissue implantation [46,47,48]. Additionally, previous findings indicated that the phenotype of the 3D lung cancer tumour sphere in vitro is closer to that of real cancer tissue in vivo [49,50]. Thus, it is considered a reasonable method to evaluate the in vivo efficacy of active compounds in the early stages of new drug development [51]. To gain insight into the effects of long-term **13k** treatment, we used a 3D spheroid tumour growth model that was built using HCC827 cancer cells. After the 3D tumour spheres had been formed, they were treated with different concentrations of **13k** for 12 days, changing the drug-containing culture medium every 3 days. As shown in Figure 8, the tumour spheres were slightly contracted and flattened after treatment with 0.4 μM **13k** for 12 days. However, the spheres were gradually split and became loose and eventually collapsed when treated with increased concentration of **13k** (0.8 μM and 1.6 μM), indicating that **13k** could effectively inhibit the tumour sphere formation and has potential for further preclinical studies.

## 3. Conclusions

In summary, a series of new 6-(imidazo[1,2-a]pyridin-6-yl)quinazoline derivatives (**10a**–**u** and **13a**–**k**) were designed, synthesized and evaluated for their in vitro anti-proliferative activities against five cancer cell lines (HCC827, A549, SH-SY5Y, HEL and MCF-7). As a result, most of the synthetic compounds showed submicromolar inhibitory activity against various tumour cell lines. Among them, **13k** is the most potent compound with IC_50_ values ranging from 0.09 μΜ to 0.43 μΜ against all the test cell lines. Moreover, compound **13k** showed strong inhibitory activity against PI3Kα, and **13k** induced cell cycle arrest at G2/M phase and cell apoptosis of HCC827 cells by inhibition of PI3Kα with an IC_50_ value of 1.94 nM. Compound **13k** showed better antitumour activity and PI3Kα kinase activity compared to the lead compound **6b**. Therefore, compound **13k** could be a promising PI3Kα inhibitor for the development of novel targeted antitumour drugs.

## 4. Experimental Procedure

### 4.1. Chemistry

#### 4.1.1. Instruments and Materials

All reagents and solvents were commercially available and used without further purification. ^1^H NMR, ^13^C NMR and ^19^F NMR spectra were recorded with a 600, 150 and 565MHz NMR spectrometer (Bruker AVANCE NEO), respectively. The NMR spectra were generated by using Mestrenova 12.0 as processing software, deuterated chloroform (CDCl_3_) and dimethyl sulfoxide-*d_6_* (DMSO-*d_6_*) as solvents, and tetramethylsilane (TMS) as an internal standard. All chemical shifts are expressed in ppm (*δ*), and the coupling constants (*J*) are expressed in hertz (Hz). The melting points of the compounds were determined using a Beijing micro melting point apparatus. High-resolution accurate mass measurements were performed on a quadrupole time-of-flight (QTOF) mass spectrometer (micro TOF-Q, Bruker Inc., Billerica, MA, USA) using electrospray ionisation (positive mode).

#### 4.1.2. General Experimental Protocol for Preparation of Compounds **10a**–**u**

##### Preparation of 4-Chloro-6-iodoquinazoline (**2**)

A mixture of 6-iodoquinazolin-4(3H)-one (2.45 g, 9 mmol), N, N-diisopropylethylamine (2.33 g, 18 mmol), phosphorus oxychloride (2.76 g, 18 mmol) and anhydrous toluene (50 mL) was reacted at 80 °C for 4 h under argon atmosphere. After completion of the reaction (monitored by TLC), the crude reaction mixture was cooled, and the solvent was removed under reduced pressure. The mixture was extracted 2–3 times with ethyl acetate and saturated sodium bicarbonate solution. The organic phase was dried with anhydrous Na_2_SO_4_ and rotary dried under vacuum. The residue was purified through a column chromatography on silica with EtOAc/PE to afford 4-chloro-6-iodoquinazoline 2 as white flocculent (2.27 g, 7.81 mmol, 86.83% yield), ESI-MS: *m*/*z* 291.5 [M + H]^+^.

##### Steps for the Preparation of 6-Iodo-N-(4-methoxybenzyl)quinazolin-4-amine (**5a**)

A mixture of 4-chloro-6-iodoquinazoline (0.58 g, 2 mmol) and 4-Methoxybenzylamine (0.33 g, 2.4 mmol) was added to isopropanol (10 mL) and refluxed at 60 °C for 2 h under argon protection. After completion of the reaction (monitored by TLC), the solvent of the reaction mixture was removed under reduced pressure and extracted 2–3 times with ethyl acetate and saturated Na_2_CO_3_ solution. The organic phase was dried over anhydrous Na_2_SO_4_ and rotary dried under vacuum to form 6-iodo-N-(4-methoxybenzyl)quinazolin-4-amine **5a** as white solid (0.60 g, 1.54 mmol, 77.0% yield), ESI-MS: *m*/*z* 392.1 [M + H]^+^.

Compounds **5b**–**q** were synthesized according to the procedure described in **5a**. The ESI-MS information of compounds **5b**–**q** is listed as below:

##### N-(cyclopropylmethyl)-6-iodoquinazolin-4-amine (**5b**)

Off-white solid, 91.2% yield, ESI-MS: *m*/*z* 326.0 [M + H]^+^.

##### N-(4-fluorobenzyl)-6-iodoquinazolin-4-amine (**5c**)

Off-white solid, 77.8% yield, ESI-MS: *m*/*z* 380.0 [M + H]^+^.

##### 6-iodo-N-(4-(trifluoromethyl)benzyl)quinazolin-4-amine (**5d**)

Off-white solid, 86.1% yield, ESI-MS: *m*/*z* 430.2 [M + H]^+^.

##### 6-iodo-N-(3-methylbenzyl)quinazolin-4-amine (**5e**)

Pale yellow solid, 94.5% yield, ESI-MS: *m*/*z* 376.2 [M + H]^+^.

##### N^1^,N^1^-diethyl-N^2^-(6-iodoquinazolin-4-yl)ethane-1,2-diamine (**5f**)

Pale yellow oily substance, 91.0% yield, ESI-MS: *m*/*z* 371.0 [M + H]^+^.

##### 6-iodo-N-(2-methylbenzyl)quinazolin-4-amine (**5g**)

Pale yellow solid, 95.3% yield, ESI-MS: *m*/*z* 376.0 [M + H]^+^.

##### 6-iodo-N-(pyridin-2-ylmethyl)quinazolin-4-amine (**5h**)

Pale yellow solid, 92.1% yield, ESI-MS: *m*/*z* 385.2 [M + Na]^+^.

##### N-(2-fluorobenzyl)-6-iodoquinazolin-4-amine (**5i**)

Off-white solid, 61.9% yield, ESI-MS: *m*/*z* 380.0 [M + H]^+^.

##### N-(3-fluorophenyl)-6-iodoquinazolin-4-amine (**5j**)

Pale yellow solid, 85.3% yield, ESI-MS: *m*/*z* 366.1 [M + H]^+^.

##### N-(3,5-dimethoxyphenyl)-6-iodoquinazolin-4-amine (**5k**)

Pale yellow solid, 90.7% yield, ESI-MS: *m*/*z* 408.2 [M + H]^+^.

##### 6-iodo-N-(pyridin-3-ylmethyl)quinazolin-4-amine (**5l**)

Pink solid, 94.2% yield, ESI-MS: *m*/*z* 385.2 [M + H]^+^.

##### N-(2,3-difluorophenyl)-6-iodoquinazolin-4-amine (**5m**)

Off-white solid, 93.6% yield, ESI-MS: *m*/*z* 384.1 [M + H]^+^.

##### 6-iodo-N-methyl-N-(p-tolyl)quinazolin-4-amine (**5n**)

Pale yellow solid, 94.0% yield, ESI-MS: *m*/*z* 376.0 [M + H]^+^.

##### N-ethyl-6-iodo-N-phenylquinazolin-4-amine (**5o**)

Pale yellow solid, 95.7% yield, ESI-MS: *m*/*z* 376.0 [M + H]^+^.

##### 6-iodo-N-(1H-pyrazol-3-yl)quinazolin-4-amine (**5p**)

White solid, 91.5% yield, ESI-MS: *m*/*z* 338.1 [M + H]^+^.

##### 6-iodo-N-((tetrahydro-2H-pyran-4-yl)methyl)quinazolin-4-amine (**5q**)

White solid, 90.1% yield, ESI-MS: *m*/*z* 392.0 [M + Na]^+^.

##### Procedure for the Preparation of 6-(6-Aminopyridin-3-yl)-N-(4-methoxybenzyl)quinazolin-4-amine (**7a**)

The 6-iodo-N-(4-methoxybenzyl)quinazolin-4-amine **5a** (0.6 g, 1.5 mmol), 5-(4,4,5,5-tetramethyl-1,3,2-dioxaborolan-2-yl)pyridin-2-amine (0.34 g, 1.5 mmol) and K_2_CO_3_ (0.64 g, 4.6 mmol) were added to 15 mL of solvent [V_(1,4-dioxane)_:V_(water)_ = 4:1]. The mixture was heated to 100 °C under a protective atmosphere of argon followed by the addition of Pd(dppf)Cl_2_. The mixture continues to be stirred under these conditions for a further 4–6 h. After completion of the reaction (monitored by TLC), 1,4-dioxane and water were removed under reduced pressure, and the residue was purified through a column chromatography on silica with dichloromethane/methanol to afford white solid 6-(6-aminopyridin-3-yl)-N-(4-methoxybenzyl)quinazolin-4-amine **7a** (0.36 g, 0.99 mmol, 66.6% yield), ESI-MS: *m*/*z* 358.1 [M + H]^+^.

Compounds **7b**–**o** were synthesized according to the procedure described in **7a**. The ESI-MS information of compounds **7b**–**o** is listed as below:

##### 6-(6-Aminopyridin-3-yl)-N-(cyclopropylmethyl)quinazolin-4-amine (**7b**)

Off-white solid, 71.2% yield, ESI-MS: *m*/*z* 291.1 [M + H]^+^.

##### 6-(6-Aminopyridin-3-yl)-N-(4-fluorobenzyl)quinazolin-4-amine (**7c**)

Off-white solid, 92.2% yield, ESI-MS: *m*/*z* 246.1 [M + H]^+^.

##### 6-(6-Aminopyridin-3-yl)-N-(4-(trifluoromethyl)benzyl)quinazolin-4-amine (**7d**)

Off-white solid, 85.7% yield, ESI-MS: *m*/*z* 396.1 [M + H]^+^.

##### 6-(6-Aminopyridin-3-yl)-N-(3-methylbenzyl)quinazolin-4-amine (**7e**)

Off-white solid, 73.4% yield, ESI-MS: *m*/*z* 342.1 [M + H]^+^.

##### N1-(6-(6-aminopyridin-3-yl)quinazolin-4-yl)-N2,N2-diethylethane-1,2-diamine (**7f**)

Brown solid, 82.9% yield, ESI-MS: *m*/*z* 359.1 [M + Na]^+^.

##### 6-(6-Aminopyridin-3-yl)-N-(2-methylbenzyl)quinazolin-4-amine (**7g**)

Off-white solid, 76.3% yield, ESI-MS: *m*/*z* 342.1 [M + H]^+^.

##### 6-(6-Aminopyridin-3-yl)-N-(pyridin-2-ylmethyl)quinazolin-4-amine (**7h**)

Yellow solid, 70.6% yield, ESI-MS: *m*/*z* 328.1 [M + H]^+^.

##### 6-(6-Aminopyridin-3-yl)-N-(2-fluorobenzyl)quinazolin-4-amine (**7i**)

Off-white solid, 86.3% yield, ESI-MS: *m*/*z* 346.1 [M + H]^+^.

##### 6-(6-Aminopyridin-3-yl)-N-(3-fluorophenyl)quinazolin-4-amine (**7j**)

Pale yellow solid, 77.6% yield, ESI-MS: *m*/*z* 332.1 [M + H]^+^.

##### 6-(6-Aminopyridin-3-yl)-N-(3,5-dimethoxyphenyl)quinazolin-4-amine (**7k**)

Yellow solid, 81.3% yield, ESI-MS: *m*/*z* 396.1 [M + Na]^+^.

##### 6-(6-Aminopyridin-3-yl)-N-(pyridin-3-ylmethyl)quinazolin-4-amine (**7l**)

Off-white solid, 67.6% yield, ESI-MS: *m*/*z* 329.1 [M + H]^+^.

##### 6-(6-Aminopyridin-3-yl)-N-(2,3-difluorophenyl)quinazolin-4-amine (**7m**)

White solid, 88.7% yield, ESI-MS: *m*/*z* 352.1 [M + Na]^+^.

##### 6-(6-Aminopyridin-3-yl)-N-methyl-N-(p-tolyl)quinazolin-4-amine (**7n**)

Pale yellow solid, 79.8% yield, ESI-MS: *m*/*z* 342.1 [M + H]^+^.

##### 6-(6-Aminopyridin-3-yl)-N-ethyl-N-phenylquinazolin-4-amine (**7o**)

Yellow solid, 75.7% yield, ESI-MS: *m*/*z* 342.1 [M + H]^+^.

##### Procedure for the Preparation of Ethyl 6-(4-((4-Methoxybenzyl)amino)quinazolin-6-yl)imidazo[1,2-a]pyridine-2-carboxylate (**10a**) or Methyl 6-(4-((4-methoxybenzyl)amino)quinazolin-6-yl)imidazo[1,2-a]pyridine-2-carboxylate (**10o**)

A mixture of 6-(6-aminopyridin-3-yl)-N-(4-methoxybenzyl)quinazolin-4-amine **7a** (0.18 g, 0.5 mmol), ethyl bromopyruvate (0.29 g, 1.5 mmol) or methyl bromopyruvate (0.27 g, 1.5 mmol) and NaHCO_3_ (0.13 g, 1.5 mmol) was added to EtOH (5 mL), and the mixture was warmed to 80 °C and refluxed by condensation under argon for 4 h. After completion of the reaction (monitored by TLC), the solvents were removed under reduced pressure and the residue was purified through a column chromatography on silica with dichloromethane/methanol to obtain ethyl 6-(4-((4-methoxybenzyl)amino)quinazolin-6-yl)imidazo[1,2-a]pyridine-2-carboxylate **10a** or methyl 6-(4-((4-methoxybenzyl)amino)quinazolin-6-yl)imidazo[1,2-a]pyridine-2-carboxylate **10o**; **10a** as white solid (0.142 g, 0.31 mmol, 62.0% yield), m.p. 131.2–133.6 °C. ^1^H NMR (600 MHz, Chloroform-d) δ 8.69 (s, 1H), 8.22 (s, 1H), 8.09 (s, 1H), 8.05 (s, 1H), 7.89 (d, *J* = 8.6 Hz, 1H), 7.83 (d, *J* = 8.5 Hz, 1H), 7.53 (d, *J* = 9.0 Hz, 1H), 7.40 (d, *J* = 9.4 Hz, 1H), 7.33 (d, *J* = 8.1 Hz, 2H), 7.24–7.17 (m, 1H), 6.83 (d, *J* = 8.0 Hz, 2H), 4.84 (s, 2H), 4.37 (q, *J* = 7.1 Hz, 2H), 3.76 (s, 3H), 1.37 (t, *J* = 7.1 Hz, 3H). ^13^C NMR (150 MHz, CDCl_3_) δ 163.1, 159.5, 159.2, 156.0, 149.1, 144.3, 137.3, 134.1, 131.2, 130.2, 129.6 (2C), 129.2, 127.5, 126.9, 123.6, 120.0, 118.7, 117.3, 115.4, 114.1 (2C), 61.3, 55.3, 44.9, 14.4. HRMS (ESI): calcd for C_26_H_24_O_3_N_5_ [M + H]^+^
*m*/*z* 454.1874, found 454.1866; C_26_H_23_O_3_N_5_Na [M + Na]^+^
*m*/*z* 476.1693, found 476.1687; **10o** as white solid (0.096 g, 0.218 mmol, 43.7% yield), m.p. 138.6–140.9 °C. ^1^H NMR (600 MHz, DMSO-*d_6_*) δ 9.25 (s, 1H), 9.04 (d, *J* = 0.8 Hz, 1H), 8.73 (s, 1H), 8.54 (s, 2H), 8.13 (d, *J* = 8.7 Hz, 1H), 7.86 (d, *J* = 9.6 Hz, 1H), 7.80 (d, *J* = 8.6 Hz, 1H), 7.75 (d, *J* = 9.7 Hz, 1H), 7.34 (d, *J* = 8.7 Hz, 2H), 6.89 (d, *J* = 8.7 Hz, 2H), 4.77 (d, *J* = 5.7 Hz, 2H), 3.85 (s, 3H), 3.71 (s, 3H). ^13^C NMR (150 MHz, DMSO) δ 163.0, 159.5, 158.4, 154.9, 148.5, 143.9, 135.9, 133.6, 131.2, 130.9, 130.9, 128.9 (2C), 126.8, 125.5, 125.1, 120.8, 118.5, 118.0, 115.0, 113.8 (2C), 55.1, 51.7, 43.4. HRMS (ESI): calcd for C_25_H_22_O_3_N_5_ [M + H]^+^
*m*/*z* 440.1717, found 440.1711.

Compounds **10b**–**n** and **10p**–**u** were synthesized according to the procedure described in **10a** or **10o**. The information of compounds **10b**–**n** and **10p**–**u** is listed as below:

##### Ethyl 6-(4-((Cyclopropylmethyl)amino)quinazolin-6-yl)imidazo[1,2-a]pyridine-2-carboxylate (**10b**)

White solid, 49.8% yield, m.p. 128.5–130.8 °C. ^1^H NMR (600 MHz, DMSO-*d*_6_) δ 9.03 (s, 1H), 8.64 (d, *J* = 2.1 Hz, 1H), 8.57 (d, *J* = 4.9 Hz, 2H), 8.47 (s, 1H), 8.09 (dd, *J* = 8.7, 2.0 Hz, 1H), 7.87 (dd, *J* = 9.6, 1.9 Hz, 1H), 7.79 (dd, *J* = 9.1, 6.6 Hz, 2H), 4.33 (q, *J* = 7.1 Hz, 2H), 3.48–3.42 (m, 2H), 1.33 (t, *J* = 7.1 Hz, 3H), 1.22 (ddd, *J* = 11.6, 7.3, 5.3 Hz, 1H), 0.53–0.46 (m, 2H), 0.34–0.29 (m, 2H). ^13^C NMR (150 MHz, DMSO) δ 162.6, 159.5, 155.5, 148.9, 143.8, 136.2, 133.2, 130.7, 128.4, 126.9, 125.8, 125.0, 120.7, 118.4, 118.0, 115.2, 60.3, 45.1, 14.3, 10.6, 3.6 (2C). HRMS (ESI): calcd for C_22_H_22_O_2_N_5_ [M + H]^+^
*m*/*z* 388.1768, found 388.1760; C_22_H_21_O_2_N_5_Na [M + Na]^+^
*m*/*z* 410.1588, found 410.1580.

##### Ethyl 6-(4-((4-Fluorobenzyl)amino)quinazolin-6-yl)imidazo[1,2-a]pyridine-2-carboxylate (**10c**)

White solid, 55.1% yield, m.p. 127.8–129.1 °C. ^1^H NMR (600 MHz, DMSO-*d*_6_) δ 9.04 (t, *J* = 1.3 Hz, 1H), 9.01 (t, *J* = 5.9 Hz, 1H), 8.68 (d, *J* = 2.2 Hz, 1H), 8.56 (s, 1H), 8.49 (s, 1H), 8.12 (dd, *J* = 8.7, 2.0 Hz, 1H), 7.87 (dd, *J* = 9.6, 1.9 Hz, 1H), 7.82 (d, *J* = 8.7 Hz, 1H), 7.79 (d, *J* = 9.5 Hz, 1H), 7.45 (dd, *J* = 8.5, 5.7 Hz, 2H), 7.16 (t, *J* = 8.9 Hz, 2H), 4.81 (d, *J* = 5.7 Hz, 2H), 4.33 (q, *J* = 7.1 Hz, 2H), 1.33 (t, *J* = 7.1 Hz, 3H). ^13^C NMR (150 MHz, DMSO) δ 162.6, 162.1, 160.5, 159.4, 155.4, 148.9, 143.9, 136.2, 135.5, 133.4, 130.9, 129.4, 129.4, 128.5, 126.8, 125.7, 125.0, 120.7, 118.4, 118.0, 115.2, 115.0, 60.3, 43.0, 14.3. ^19^F NMR (565 MHz, DMSO) δ −115.99. HRMS (ESI): calcd for C_25_H_21_O_2_N_5_F [M + H]^+^
*m*/*z* 442.1674, found 442.1667; C_25_H_20_O_2_N_5_FNa [M + Na]^+^
*m*/*z* 464.1493, found 464.1489.

##### Ethyl 6-(4-((4-(Trifluoromethyl)benzyl)amino)quinazolin-6-yl)imidazo[1,2-a]pyridine-2-carboxylate (**10d**)

White solid, 56,8% yield, m.p. 129.8–131.1 °C. ^1^H NMR (600 MHz, DMSO-*d*_6_) δ 9.14 (t, *J* = 6.0 Hz, 1H), 9.05 (s, 1H), 8.70 (s, 1H), 8.56 (s, 1H), 8.49 (s, 1H), 8.14 (d, *J* = 8.5 Hz, 1H), 7.87 (d, *J* = 9.4 Hz, 1H), 7.84 (d, *J* = 8.6 Hz, 1H), 7.80 (d, *J* = 9.4 Hz, 1H), 7.70 (d, *J* = 8.0 Hz, 2H), 7.61 (d, *J* = 8.1 Hz, 2H), 4.92 (d, *J* = 5.6 Hz, 2H), 4.33 (q, *J* = 7.1 Hz, 2H), 1.33 (t, *J* = 7.1 Hz, 3H). ^13^C NMR (150 MHz, DMSO) δ 162.6, 159.5, 155.3, 148.7, 144.3, 143.8, 136.2, 133.5, 131.0, 128.4, 128.0 (2C), 127.7, 126.7, 125.6, 125.3 (2C), 125.0, 123.5, 120.7, 118.4, 118.0, 115.1, 60.3, 43.4, 14.3. ^19^F NMR (565 MHz, DMSO) δ −60.78. HRMS (ESI): calcd for C_26_H_21_O_2_N_5_F_3_ [M + H]^+^
*m*/*z* 492.1642, found 492.1632; C_26_H_20_O_2_N_5_F_3_Na [M + Na]^+^
*m*/*z* 514.1461, found 514.1452.

##### Ethyl 6-(4-((3-Methylbenzyl)amino)quinazolin-6-yl)imidazo[1,2-a]pyridine-2-carboxylate (**10e**)

Off-white solid 53.0% yield, m.p. 130.1–132.5 °C. ^1^H NMR (600 MHz, DMSO-*d*_6_) δ 9.03 (s, 1H), 9.01 (t, *J* = 5.9 Hz, 1H), 8.70 (s, 1H), 8.55 (s, 1H), 8.50 (s, 1H), 8.12 (d, *J* = 7.9 Hz, 1H), 7.87 (d, *J* = 9.4 Hz, 1H), 7.82 (d, *J* = 8.6 Hz, 1H), 7.79 (d, *J* = 9.5 Hz, 1H), 7.20 (dd, *J* = 12.5, 7.3 Hz, 3H), 7.06 (d, *J* = 7.1 Hz, 1H), 4.80 (d, *J* = 5.7 Hz, 2H), 4.33 (q, *J* = 7.1 Hz, 2H), 2.28 (s, 3H), 1.33 (t, *J* = 7.1 Hz, 3H). ^13^C NMR (150 MHz, DMSO) δ 162.6, 159.5, 155.4, 148.7, 143.9, 139.2, 137.5, 136.2, 133.4, 130.9, 128.3, 128.3, 128.0, 127.6, 126.8, 125.7, 125.0, 124.5, 120.7, 118.5, 118.0, 115.2, 60.4, 43.7, 21.1, 14.3. HRMS (ESI): calcd for C_26_H_24_O_2_N_5_ [M + H]^+^
*m*/*z* 438.1925, found 438.1918; C_26_H_23_O_2_N_5_Na [M + Na]^+^
*m*/*z* 460.1744, found 460.1737.

##### Ethyl 6-(4-((2-(Diethylamino)ethyl)amino)quinazolin-6-yl)imidazo[1,2-a]pyridine-2-carboxylate (**10f**)

Brown solid, 42.1% yield, m.p. 235.7–237.9 °C. ^1^H NMR (600 MHz, DMSO-d6) δ 9.28 (s, 1H), 9.24 (s, 1H), 8.98 (s, 1H), 8.51 (d, *J* = 11.6 Hz, 2H), 8.16 (d, *J* = 8.7 Hz, 1H), 8.06 (d, *J* = 9.6 Hz, 1H), 7.81 (d, *J* = 8.6 Hz, 1H), 7.74 (d, *J* = 9.4 Hz, 1H), 4.33 (q, *J* = 7.1 Hz, 2H), 3.94 (s, 2H), 3.17 (s, 6H), 1.33 (t, *J* = 7.1 Hz, 3H), 1.22 (s, 6H). ^13^C NMR (150 MHz, DMSO) δ 162.6, 159.6, 155.2, 148.8, 143.9, 136.1, 133.1, 130.6, 128.3, 126.7, 125.3, 125.2, 121.1, 118.3, 117.9, 115.4, 60.4 (2C), 46.6 (3C), 14.3 (3C). HRMS (ESI): calcd for C_24_H_29_O_2_N_6_ [M + H]^+^
*m*/*z* 433.2347, found 433.2341.

##### Ethyl 6-(4-((2-Methylbenzyl)amino)quinazolin-6-yl)imidazo[1,2-a]pyridine-2-carboxylate (**10g**)

Pink solid, 45.8% yield, m.p. 149.6–151.7 °C. ^1^H NMR (600 MHz, DMSO-d6) δ 9.05 (s, 1H), 8.94 (s, 1H), 8.76 (s, 1H), 8.56 (s, 1H), 8.51 (s, 1H), 8.15 (d, *J* = 8.6 Hz, 1H), 7.88 (d, *J* = 9.4 Hz, 1H), 7.84 (d, *J* = 8.5 Hz, 1H), 7.80 (d, *J* = 9.5 Hz, 1H), 7.30 (d, *J* = 7.3 Hz, 1H), 7.22–7.14 (m, 3H), 4.81 (d, *J* = 5.4 Hz, 2H), 4.33 (q, *J* = 7.1 Hz, 2H), 2.37 (s, 3H), 1.33 (t, *J* = 7.0 Hz, 3H). ^13^C NMR (150 MHz, DMSO) δ 162.6, 159.5, 155.2, 143.8, 136.5, 136.2, 135.9, 133.5, 133.5, 131.0, 130.0, 128.0, 127.4, 127.0, 126.8, 125.8, 125.7, 125.0, 120.8, 118.4, 118.0, 115.1, 60.3, 42.1, 18.8, 14.3. HRMS (ESI): calcd for C_26_H_24_O_2_N_5_ [M + H]^+^
*m*/*z* 438.1925, found 438.1917; C_26_H_23_O_2_N_5_Na [M + Na]^+^
*m*/*z* 460.1744, found 460.1735.

##### Ethyl 6-(4-((Pyridin-2-ylmethyl)amino)quinazolin-6-yl)imidazo[1,2-a]pyridine-2-carboxylate (**10h**)

Yellow solid, 57.3% yield, m.p. 129.6–131.5 °C. ^1^H NMR (600 MHz, DMSO-*d*_6_) δ 9.20–9.14 (m, 1H), 9.05 (s, 1H), 8.73 (s, 1H), 8.58–8.51 (m, 2H), 8.47 (s, 1H), 8.14 (d, *J* = 8.6 Hz, 1H), 7.88 (d, *J* = 9.5 Hz, 1H), 7.84–7.78 (m, 2H), 7.73 (t, *J* = 7.6 Hz, 1H), 7.39 (d, *J* = 7.9 Hz, 1H), 7.29–7.24 (m, 1H), 4.92 (d, *J* = 5.7 Hz, 2H), 4.32 (q, *J* = 7.1 Hz, 2H), 1.33 (t, *J* = 7.1 Hz, 3H). ^13^C NMR (150 MHz, DMSO) δ 162.6, 159.6, 158.5, 155.3, 149.0, 148.6, 143.8, 136.8, 136.2, 133.4, 130.9, 128.3, 126.7, 125.6, 125.0, 122.2, 121.2, 120.7, 118.4, 118.0, 115.2, 60.3, 45.7, 14.3. HRMS (ESI): calcd for C_24_H_21_O_2_N_6_ [M + H]^+^
*m*/*z* 425.1721, found 425.1712; C_24_H_20_O_2_N_6_Na [M + Na]^+^
*m*/*z* 447.1540, found 447.1532.

##### Ethyl 6-(4-((2-Fluorobenzyl)amino)quinazolin-6-yl)imidazo[1,2-a]pyridine-2-carboxylate (**10i**)

Off-white solid, 59.2% yield, m.p. 142.3–144.6 °C. ^1^H NMR (600 MHz, DMSO-*d*_6_) δ 9.48 (s, 1H), 9.08 (s, 1H), 8.81 (s, 1H), 8.60 (s, 1H), 8.55(s, 1H), 8.19 (d, *J* = 8.4 Hz, 1H), 7.88 (d, *J* = 9.4 Hz, 1H), 7.85 (d, *J* = 8.4 Hz, 1H), 7.78 (d, *J* = 9.4 Hz, 1H), 7.46 (t, *J* = 7.6 Hz, 1H), 7.33 (q, *J* = 6.9 Hz, 1H), 7.22 (t, *J* = 9.3 Hz, 1H), 7.16 (t, *J* = 7.4 Hz, 1H), 4.90 (d, *J* = 5.2 Hz, 2H), 4.33 (q, *J* = 7.0 Hz, 2H), 1.33 (t, *J* = 7.0 Hz, 3H). ^13^C NMR (150 MHz, DMSO) δ 162.5, 161.1, 159.8, 159.5, 154.3, 143.8, 136.2, 134.1, 131.6, 129.6, 129.1, 126.7, 126.5, 125.3, 125.2, 124.4, 121.0, 118.4, 118.0, 115.3, 115.2, 114.7, 60.3, 38.0, 14.3. ^19^F NMR (565 MHz, DMSO) δ −118.51. HRMS (ESI): calcd for C_25_H_21_O_2_N_5_F [M + H]^+^
*m*/*z* 442.1674, found 442.1664; C_25_H_20_O_2_N_5_FNa [M + Na]^+^
*m*/*z* 464.1493, found 464.1486.

##### Ethyl 6-(4-((3-Fluorophenyl)amino)quinazolin-6-yl)imidazo[1,2-a]pyridine-2-carboxylate (**10j**)

Yellow solid, 46.7% yield, m.p. 148.3–150.1 °C. ^1^H NMR (600 MHz, DMSO-*d*_6_) δ 10.09 (s, 1H), 9.08 (s, 1H), 8.86 (s, 1H), 8.68 (s, 1H), 8.56 (s, 1H), 8.19 (d, *J* = 8.7 Hz, 1H), 7.92 (t, *J* = 8.0 Hz, 3H), 7.82 (d, *J* = 9.4 Hz, 1H), 7.68 (d, *J* = 8.2 Hz, 1H), 7.45 (q, *J* = 7.8 Hz, 1H), 6.98 (td, *J* = 8.5, 2.6 Hz, 1H), 4.33 (q, *J* = 7.1 Hz, 2H), 1.33 (t, *J* = 7.1 Hz, 3H). ^13^C NMR (150 MHz, DMSO) δ 162.9, 162.6, 161.3, 157.7, 154.6, 149.2, 143.9, 136.3, 134.3, 131.6, 130.1, 128.7, 126.9, 125.6, 125.4, 120.8, 118.5, 118.1, 117.9, 115.4, 110.4, 109.1, 60.4, 14.3. ^19^F NMR (565 MHz, DMSO) δ −112.46. HRMS (ESI): calcd for C_24_H_18_O_2_N_5_FNa [M + Na]^+^
*m*/*z* 450.1337, found 450.1328.

##### Ethyl 6-(4-((3,5-dimethoxyphenyl)amino)quinazolin-6-yl)imidazo[1,2-a]pyridine-2-carboxylate (**10k**)

Yellow solid, 50.2% yield, m.p. 156.2–158.3 °C. ^1^H NMR (600 MHz, DMSO-*d*_6_) δ 9.90 (s, 1H), 9.09 (s, 1H), 8.87 (s, 1H), 8.66 (s, 1H), 8.57 (s, 1H), 8.19 (d, *J* = 8.4 Hz, 1H), 7.94 (d, *J* = 9.3 Hz, 1H), 7.90 (d, *J* = 8.6 Hz, 1H), 7.82 (d, *J* = 9.4 Hz, 1H), 7.19 (s, 2H), 6.34 (s, 1H), 4.33 (q, *J* = 7.1 Hz, 2H), 3.78 (s, 6H), 1.34 (t, *J* = 7.1 Hz, 3H). ^13^C NMR (150 MHz, DMSO) δ 162.6, 160.4 (2C), 157.8, 154.6, 148.9, 143.9, 140.6, 136.2, 134.2, 131.5, 128.4, 126.9, 125.6, 125.3, 120.8, 118.5, 118.0, 115.4, 100.8 (2C), 95.8, 60.3, 55.3 (2C), 14.3. HRMS (ESI): calcd for C_26_H_24_O_4_N_5_ [M + H]^+^
*m*/*z* 470.1823, found 470.1809; C_26_H_23_O_4_N_5_Na [M + Na]^+^
*m*/*z* 492.1642, found 492.1633.

##### Ethyl 6-(4-((2,3-difluorophenyl)amino)quinazolin-6-yl)imidazo[1,2-a]pyridine-2-carboxylate (**10l**)

White solid, 55.3% yield, m.p. 131.2–133.6 °C. ^1^H NMR (600 MHz, DMSO-*d*_6_) δ 10.21 (s, 1H), 9.10 (s, 1H), 8.83 (s, 1H), 8.57 (d, *J* = 7.1 Hz, 2H), 8.23 (d, *J* = 8.7 Hz, 1H), 8.02–7.86 (m, 2H), 7.82 (d, *J* = 9.5 Hz, 1H), 7.39 (s, 2H), 7.30 (s, 1H), 4.33 (q, *J* = 7.1 Hz, 2H), 1.33 (t, *J* = 7.1 Hz, 3H). ^13^C NMR (150 MHz, DMSO) δ 162.6, 158.4, 154.9, 151.3, 151.2, 149.6, 149.6, 143.9, 136.3, 134.2, 131.6, 128.6, 126.7, 125.5, 125.3, 124.3, 123.2, 121.0, 118.5, 118.1, 115.1, 114.5, 60.4, 14.3. ^19^F NMR (565 MHz, DMSO) δ −138.40, −142.37. HRMS (ESI): calcd for C_24_H_18_O_2_N_5_F_2_ [M + H]^+^
*m*/*z* 446.1423, found 446.1411; C_24_H_17_O_2_N_5_F_2_Na [M + Na]^+^
*m*/*z* 468.1243, found 468.1231.

##### Ethyl 6-(4-(Ethyl(phenyl)amino)quinazolin-6-yl)imidazo[1,2-a]pyridine-2-carboxylate (**10m**)

Pink flocculent, 63.1% yield, m.p. 176.7–177.8 °C. ^1^H NMR (600 MHz, DMSO-*d*_6_) δ 8.74 (s, 1H), 8.50 (s, 1H), 8.45 (s, 1H), 7.97 (dd, *J* = 8.7, 2.1 Hz, 1H), 7.85 (d, *J* = 8.6 Hz, 1H), 7.60–7.55 (m, 3H), 7.50 (t, *J* = 7.4 Hz, 1H), 7.38 (d, *J* = 7.6 Hz, 2H), 7.10 (d, *J* = 2.1 Hz, 1H), 6.86 (dd, *J* = 9.5, 1.8 Hz, 1H), 4.32 (q, *J* = 7.1 Hz, 2H), 4.20 (q, *J* = 6.9 Hz, 2H), 1.33 (t, *J* = 7.1 Hz, 3H), 1.25 (t, *J* = 7.0 Hz, 3H). ^13^C NMR (150 MHz, DMSO) δ 162.5, 160.0, 154.5, 151.1, 145.8, 143.6, 136.2, 131.9, 130.5 (2C), 130.4, 129.1, 127.3, 127.2 (2C), 125.8, 125.4, 124.7, 124.0, 118.4, 117.9, 115.8, 60.4, 48.2, 14.3, 11.7. HRMS (ESI): calcd for C_26_H_24_O_2_N_5_ [M + H]^+^
*m*/*z* 438.1925, found 438.1916; C_26_H_23_O_2_N_5_Na [M + Na]^+^
*m*/*z* 460.1744, found 460.1735.

##### Ethyl 6-(4-((Pyridin-3-ylmethyl)amino)quinazolin-6-yl)imidazo[1,2-a]pyridine-2-carboxylate (**10n**)

White solid, 47.3% yield, m.p. 120.3–122.4 °C. ^1^H NMR (600 MHz, DMSO-*d*_6_) δ 9.08 (t, *J* = 5.9 Hz, 1H), 9.04 (s, 1H), 8.66 (d, *J* = 7.1 Hz, 2H), 8.55 (s, 1H), 8.51 (s, 1H), 8.47 (d, *J* = 4.8 Hz, 1H), 8.13 (d, *J* = 8.6 Hz, 1H), 7.87 (d, *J* = 9.5 Hz, 1H), 7.84–7.77 (m, 3H), 7.36 (dd, *J* = 7.9, 4.7 Hz, 1H), 4.85 (d, *J* = 5.7 Hz, 2H), 4.33 (q, *J* = 7.1 Hz, 2H), 1.33 (t, *J* = 7.1 Hz, 3H). ^13^C NMR (150 MHz, DMSO) δ 162.6, 159.5, 155.3, 149.0, 148.7, 148.2, 143.8, 136.2, 135.3, 134.8, 133.5, 131.0, 128.4, 126.8, 125.7, 125.0, 123.6, 120.7, 118.4, 118.0, 115.1, 60.4, 41.5, 14.3. HRMS (ESI): calcd for C_24_H_21_O_2_N_6_ [M + H]^+^
*m*/*z* 425.1721, found 425.1711; C_24_H_20_O_2_N_6_Na [M + Na]^+^
*m*/*z* 447.1540, found 447.1534.

##### Methyl 6-(4-((2-Methylbenzyl)amino)quinazolin-6-yl)imidazo[1,2-a]pyridine-2-carboxylate (**10p**)

Off-white solid, 36.7% yield, m.p. 150.3–152.5 °C. ^1^H NMR (600 MHz, DMSO-*d*_6_) δ 9.05 (s, 2H), 8.77 (s, 1H), 8.57 (s, 1H), 8.53 (s, 1H), 8.23–8.08 (m, 1H), 7.96–7.70 (m, 3H), 7.30 (d, *J* = 7.2 Hz, 1H), 7.24–7.11 (m, 3H), 4.81 (d, *J* = 4.7 Hz, 2H), 3.86 (s, 3H), 2.37 (s, 3H). ^13^C NMR (150 MHz, DMSO) δ 163.0, 159.6, 155.0, 147.7, 143.9, 136.4, 135.9, 133.7, 131.7, 131.2, 130.1, 127.6, 127.5, 127.0, 126.9, 125.8, 125.6, 125.1, 120.9, 118.5, 118.0, 115.0, 51.7, 42.2, 18.9. HRMS (ESI): calcd for C_25_H_22_O_2_N_5_ [M + H]^+^
*m*/*z* 424.1768, found 424.1760; C_25_H_21_O_2_N_5_Na [M + Na]^+^
*m*/*z* 446.1588, found 446.1580.

##### Methyl 6-(4-((Pyridin-2-ylmethyl)amino)quinazolin-6-yl)imidazo[1,2-a]pyridine-2-carboxylate (**10q**)

Yellow solid, 53.8% yield, m.p. 143.7–145.6 °C. ^1^H NMR (600 MHz, DMSO-*d*_6_) δ 9.23 (s, 1H), 9.06 (s, 1H), 8.75 (s, 1H), 8.57 (s, 1H), 8.53 (d, *J* = 4.8 Hz, 1H), 8.48 (s, 1H), 8.16 (d, *J* = 8.6 Hz, 1H), 7.89 (d, *J* = 9.5 Hz, 1H), 7.83 (d, *J* = 8.6 Hz, 1H), 7.80 (d, *J* = 9.5 Hz, 1H), 7.74 (t, *J* = 7.5 Hz, 1H), 7.39 (d, *J* = 7.8 Hz, 1H), 7.29–7.25 (m, 1H), 4.92 (d, *J* = 5.5 Hz, 2H), 3.86 (s, 3H). ^13^C NMR (150 MHz, DMSO) δ 163.0, 159.7, 158.4, 155.2, 149.0, 148.2, 143.9, 136.8, 135.9, 133.5, 131.0, 128.0, 126.7, 125.6, 125.1, 122.3, 121.2, 120.8, 118.5, 118.0, 115.1, 51.7, 45.8. HRMS (ESI): calcd for C_23_H_19_O_2_N_6_ [M + H]^+^
*m*/*z* 411.1564, found 411.1556; C_23_H_18_O_2_N_6_Na [M + Na]^+^
*m*/*z* 433.1384, found 433.1372.

##### Methyl 6-(4-((2-Fluorobenzyl)amino)quinazolin-6-yl)imidazo[1,2-a]pyridine-2-carboxylate (**10r**)

Off-white solid, 52.1% yield, m.p. 174.0–176.2 °C. ^1^H NMR (600 MHz, DMSO-*d*_6_) δ 9.72 (s, 1H), 9.09 (s, 1H), 8.85 (s, 1H), 8.64 (s, 1H), 8.55 (s, 1H), 8.21 (s, 1H), 7.87 (t, *J* = 11.8 Hz, 2H), 7.76 (d, *J* = 9.2 Hz, 1H), 7.47 (t, *J* = 7.6 Hz, 1H), 7.33 (s, 1H), 7.22 (t, *J* = 9.2 Hz, 1H), 7.16 (t, *J* = 7.4 Hz, 1H), 4.92 (s, 2H), 3.85 (s, 3H). ^13^C NMR (150 MHz, DMSO) δ 162.9, 161.1, 159.9, 159.4, 153.9, 143.8, 135.9, 134.3, 131.9, 129.7, 129.2, 126.6, 125.6, 125.3, 125.2, 124.4, 121.1, 118.5, 118.0, 115.3, 115.2, 114.5, 51.7, 38.1. ^19^F NMR (565 MHz, DMSO) δ −118.44. HRMS (ESI): calcd for C_24_H_19_O_2_N_5_F [M + H]^+^
*m*/*z* 428.1517, found 428.1509; C_24_H_18_O_2_N_5_FNa [M + Na]^+^
*m*/*z* 450.1337, found 450.1330.

##### Methyl 6-(4-((3-Fluorophenyl)amino)quinazolin-6-yl)imidazo[1,2-a]pyridine-2-carboxylate (**10s**)

White solid, 48.6% yield, m.p. 179.3–181.6 °C. ^1^H NMR (600 MHz, DMSO-*d*_6_) δ 10.22–10.05 (m, 1H), 9.10 (s, 1H), 8.88 (s, 1H), 8.69 (s, 1H), 8.57 (s, 1H), 8.21 (dd, *J* = 8.7, 2.0 Hz, 1H), 7.97–7.88 (m, 3H), 7.81 (d, *J* = 9.5 Hz, 1H), 7.68 (d, *J* = 8.1 Hz, 1H), 7.49–7.41 (m, 1H), 6.98 (td, *J* = 8.5, 2.6 Hz, 1H), 3.86 (s, 3H). ^13^C NMR (150 MHz, DMSO) δ 163.0, 162.8, 161.2, 157.7, 154.4, 148.8, 143.9, 140.8, 135.9, 134.3, 131.6, 130.1, 128.4, 126.9, 125.6, 125.4, 120.8, 118.5, 118.0, 115.3, 110.4, 109.2, 51.7. ^19^F NMR (565 MHz, DMSO) δ −112.46. HRMS (ESI): calcd for C_23_H_17_O_2_N_5_F [M + H]^+^
*m*/*z* 414.1361, found 414.1347; C_23_H_16_O_2_N_5_FNa [M + Na]^+^
*m*/*z* 436.1180, found 436.1172.

##### Methyl 6-(4-(Methyl(p-tolyl)amino)quinazolin-6-yl)imidazo[1,2-a]pyridine-2-carboxylate (**10t**)

Pink flocculent, 39.5% yield, m.p. 172.8–174.3 °C. ^1^H NMR (600 MHz, DMSO-*d*_6_) δ 8.72 (s, 1H), 8.58 (s, 1H), 8.41 (s, 1H), 7.95 (dd, *J* = 8.7, 2.1 Hz, 1H), 7.83 (d, *J* = 8.6 Hz, 1H), 7.57 (d, *J* = 9.5 Hz, 1H), 7.36 (d, *J* = 8.0 Hz, 2H), 7.27 (d, *J* = 8.2 Hz, 2H), 7.06 (d, *J* = 2.1 Hz, 1H), 6.77 (dd, *J* = 9.5, 1.9 Hz, 1H), 3.85 (s, 3H), 3.57 (s, 3H), 2.41 (s, 3H). ^13^C NMR (150 MHz, DMSO) δ 162.9, 160.4, 154.4, 150.8, 145.1, 143.6, 137.0, 135.9, 131.8, 130.9 (2C), 130.2, 128.9, 126.4 (2C), 125.6, 125.3, 124.9, 124.0, 118.4, 117.8, 115.7, 51.7, 42.0, 20.6. HRMS (ESI): calcd for C_25_H_22_O_2_N_5_ [M + H]^+^
*m*/*z* 424.1768, found 424.1757; C_25_H_21_O_2_N_5_Na [M + Na]^+^
*m*/*z* 446.1588, found 446.1580.

##### Methyl 6-(4-(Ethyl(phenyl)amino)quinazolin-6-yl)imidazo[1,2-a]pyridine-2-carboxylate (**10u**)

Orange flocculent, 51.4% yield, m.p. 286.5–287.9 °C. ^1^H NMR (600 MHz, DMSO-*d*_6_) δ 8.73 (s, 1H), 8.38 (d, *J* = 9.2 Hz, 2H), 7.94 (d, *J* = 8.5 Hz, 1H), 7.84 (d, *J* = 8.6 Hz, 1H), 7.54 (d, *J* = 8.1 Hz, 3H), 7.46 (t, *J* = 7.4 Hz, 1H), 7.33 (d, *J* = 7.7 Hz, 2H), 7.20 (s, 1H), 6.98 (d, *J* = 8.7 Hz, 1H), 4.24 (q, *J* = 7.0 Hz, 2H), 3.88 (s, 3H), 1.31 (t, *J* = 6.9 Hz, 3H). ^13^C NMR (150 MHz, DMSO) δ 162.3, 159.8, 153.7 (2C), 150.7, 145.6, 143.2, 135.8, 131.5, 129.7 (2C), 129.6, 128.4, 126.4 (2C), 126.3, 125.2, 123.8, 123.4, 117.4, 117.2, 115.7, 50.7, 47.4, 11.4. HRMS (ESI): calcd for C_25_H_22_O_2_N_5_ [M + H]^+^
*m*/*z* 424.1768, found 424.1762; C_25_H_21_O_2_N_5_Na [M + Na]^+^
*m*/*z* 446.1588, found 446.1581.

#### 4.1.3. General Experimental Protocol for Preparation of Compounds **13a**–**k**

##### Procedure for the Preparation of 2-Phenyl-6-(4,4,5,5-tetramethyl-1,3,2-dioxaborolan-2-yl)imidazo[1,2-a]pyridine (**12a**)

The components 5-(4,4,5,5-Tetramethyl-1,3,2-dioxaborolan-2-yl)pyridin-2-amine (1.2 g, 5.4 mmol), 2-bromoacetophenone **11a** (1.3 g, 6.5 mmol) and NaHCO_3_ (1.4 g, 16 mmol) were added to EtOH (10 mL), and the mixture was heated to 80 °C and refluxed by condensation under argon for 4 h. After completion of the reaction (monitored by TLC), the solvent of the reaction mixture was removed under reduced pressure, and the mixture was extracted 2–3 times with ethyl acetate and saturated Na_2_CO_3_ solution. The organic phase was dried over anhydrous Na_2_SO_4_ and rotary dried under vacuum to form 2-phenyl-6-(4,4,5,5-tetramethyl-1,3,2-dioxaborolan-2-yl)imidazo[1,2-a]pyridine **12a** as pale-yellow oil substance (1.5 g, 4.7 mmol, 86.8% yield), ESI-MS: *m*/*z* 321.1 [M + H]^+^.

Compounds **12b**–**c** were synthesized according to the procedure described in **12a**. The ESI-MS information of compounds **12b**–**c** is listed as below:

##### Compound 2-(4-Fluorophenyl)-6-(4,4,5,5-tetramethyl-1,3,2-dioxaborolan-2-yl)imidazo[1,2-a]pyridine (**12b**)

Yellow solid, 81.5% yield, ESI-MS: *m*/*z* 337.2 [M + H]^+^.

##### Compound 2-Cyclopropyl-6-(4,4,5,5-tetramethyl-1,3,2-dioxaborolan-2-yl)imidazo[1,2-a]pyridine (**12c**)

Yellow solid, 81.5% yield, ESI-MS: *m*/*z* 283.2 [M + H]^+^.

##### Procedure for the Preparation of N-(2-fluorobenzyl)-6-(2-phenylimidazo[1,2-a]pyridin-6-yl)quinazolin-4-amine (**13a**)

N-(2-fluorobenzyl)-6-iodoquinazolin-4-amine 5i (0.19 g, 0.5mmol), 2-phenyl-6-(4,4,5,5-tetramethyl-1,3,2-dioxaborolan-2-yl)imidazo[1,2-a]pyridine 12a (0.16 g, 0.5 mmol) and K_2_CO_3_ (0.21 g, 1.5 mmol) were added to 1,4-dioxane/water 10 mL [V_(1,4-dioxane)_:V_(water)_ = 4:1], and the mixture was heated to 100 °C under a protective atmosphere of argon followed by the addition of Pd(dppf)Cl_2_. The mixture continues to be stirred under these conditions for a further 4–5 h. After completion of the reaction (monitored by TLC), the 1, 4-dioxane and water were removed under reduced pressure, and the residue was purified through a column chromatography on silica with dichloromethane/methanol to afford N-(2-fluorobenzyl)-6-(2-phenylimidazo[1,2-a]pyridin-6-yl)quinazolin-4-amine **13a** as pink flocculent, 67.3% yield, m.p. 130.8–132.2 °C. ^1^H NMR (600 MHz, DMSO-*d_6_*) δ 9.00 (s, 1H), 8.98 (t, *J* = 5.8 Hz, 1H), 8.72 (s, 1H), 8.50 (s, 1H), 8.44 (s, 1H), 8.18 (d, *J* = 8.7 Hz, 1H), 8.01 (d, *J* = 7.9 Hz, 2H), 7.82 (d, *J* = 8.6 Hz, 1H), 7.77 (q, *J* = 9.4 Hz, 2H), 7.46 (t, *J* = 7.6 Hz, 3H), 7.36–7.30 (m, 2H), 7.25–7.20 (m, 1H), 7.16 (t, *J* = 7.5 Hz, 1H), 4.88 (d, *J* = 5.5 Hz, 2H). ^13^C NMR (150 MHz, DMSO) δ 161.2, 159.6, 159.5, 155.2, 148.7, 145.2, 144.2, 134.0, 133.8, 131.0, 129.6, 129.1, 128.8 (2C), 128.4, 127.9, 125.9, 125.8, 125.7 (2C), 125.1, 124.4, 120.3, 116.8, 115.3, 115.2, 109.7, 37.8. ^19^F NMR (565 MHz, DMSO) δ −118.62. HRMS (ESI): calcd for C_28_H_21_N_5_F [M + H]^+^
*m*/*z* 446.1776, found 446.1768.

Compounds **13b**–**13k** were synthesized according to the procedure described in **13a**. The information of compounds **13b**–**13k** is listed as below:

##### N-(2-fluorobenzyl)-6-(2-(4-fluorophenyl)imidazo[1,2-a]pyridin-6-yl)quinazolin-4-amine (**13b**)

Pink flocculent, 70.7% yield, m.p. 133.7–135.6 °C. ^1^H NMR (600 MHz, DMSO-*d*_6_) δ 9.00 (s, 1H), 8.96 (t, *J* = 5.4 Hz, 1H), 8.72 (s, 1H), 8.50 (s, 1H), 8.42 (s, 1H), 8.18 (d, *J* = 8.7 Hz, 1H), 8.04 (dd, *J* = 8.5, 5.5 Hz, 2H), 7.82 (d, *J* = 8.6 Hz, 1H), 7.79 (d, *J* = 9.4 Hz, 1H), 7.75 (d, *J* = 9.4 Hz, 1H), 7.45 (t, *J* = 7.7 Hz, 1H), 7.35–7.31 (m, 1H) 7.29 (t, *J* = 8.8 Hz, 2H), 7.24–7.20 (m, 1H), 7.16 (t, *J* = 7.5 Hz, 1H), 4.88 (d, *J* = 5.5 Hz, 2H). ^13^C NMR (150 MHz, DMSO) δ 162.8, 161.1, 159.5, 155.2, 148.6, 144.2, 134.0, 131.0, 130.3, 129.6, 129.0, 128.3, 127.6, 125.9, 125.8, 125.1, 124.5, 124.4 (2C), 120.3, 116.7 (2C), 115.7, 115.6, 115.3, 115.2, 109.5, 37.7. ^19^F NMR (565 MHz, DMSO) δ −113.19, −117.60. HRMS (ESI): calcd for C_28_H_20_N_5_F_2_ [M + H]^+^
*m*/*z* 464.1681, found 464.1677.

##### N-(2,3-difluorophenyl)-6-(2-phenylimidazo[1,2-a]pyridin-6-yl)quinazolin-4-amine (**13c**)

Pink flocculent, 73.2% yield, m.p. 142.0–144.3 °C. ^1^H NMR (600 MHz, DMSO-*d*_6_) δ 10.22 (s, 1H), 9.08 (s, 1H), 8.86 (s, 1H), 8.56 (s, 1H), 8.46 (s, 1H), 8.28 (d, *J* = 8.4 Hz, 1H), 8.02 (d, *J* = 7.3 Hz, 2H), 7.93 (d, *J* = 8.7 Hz, 1H), 7.87–7.76 (m, 2H), 7.47 (t, *J* = 7.7 Hz, 2H), 7.40 (q, *J* = 8.6 Hz, 2H), 7.35 (t, *J* = 7.4 Hz, 1H) 7.30 (q, *J* = 7.2 Hz, 1H). ^13^C NMR (150 MHz, DMSO) δ 158.5, 154.7, 151.3, 149.7, 149.6, 145.1, 144.2, 134.7, 133.6, 131.6, 129.7, 128.8 (2C), 128.4, 128.0, 125.7 (2C), 125.0, 124.7, 124.3, 124.3, 123.3, 120.7, 116.8, 115.1, 114.6, 109.7. ^19^F NMR (565 MHz, DMSO) δ −137.37, −141.29. HRMS (ESI): calcd for C_27_H_18_N_5_F_2_ [M + H]^+^
*m*/*z* 450.1525, found 450.1520; C_27_H_17_N_5_F_2_Na [M + Na]^+^
*m*/*z* 472.1344, found 472.1337.

##### N-(2,3-difluorophenyl)-6-(2-(4-fluorophenyl)imidazo[1,2-a]pyridin-6-yl)quinazolin-4-amine (**13d**)

Pink flocculent, 72.5% yield, m.p. 136.1–138.2 °C. ^1^H NMR (600 MHz, DMSO-*d*_6_) δ 10.21 (s, 1H), 9.07 (s, 1H), 8.85 (s, 1H), 8.56 (s, 1H), 8.43 (s, 1H), 8.28 (d, *J* = 8.8 Hz, 1H), 8.05 (dd, *J* = 8.5, 5.5 Hz, 2H), 7.92 (d, *J* = 8.7 Hz, 1H), 7.84 (d, *J* = 9.4 Hz, 1H), 7.78 (d, *J* = 9.4 Hz, 1H), 7.40 (q, *J* = 8.4 Hz, 2H), 7.29 (t, *J* = 8.7 Hz, 3H). ^13^C NMR (150 MHz, DMSO) δ 162.8, 161.2, 158.4, 154.7, 151.2, 149.7, 144.2, 134.7, 131.6, 130.2, 128.4, 127.7 (2C), 127.6, 125.1, 124.7, 124.3, 124.2, 123.3, 120.6, 116.8, 115.7 (2C), 115.6, 115.1, 114.6, 109.6. ^19^F NMR (565 MHz, DMSO) δ −113.15, −137.34, −141.32. HRMS (ESI): calcd for C_27_H_17_N_5_F_3_ [M + H]^+^
*m*/*z* 468.1431, found 468.1426; C_27_H_16_N_5_F_3_Na [M + Na]^+^
*m*/*z* 490.1250, found 490.1239.

##### N-(cyclopropylmethyl)-6-(2-(4-fluorophenyl)imidazo[1,2-a]pyridin-6-yl)quinazolin-4-amine (**13e**)

Pink flocculent, 56.1% yield, m.p. 227.5–229.7 °C. ^1^H NMR (600 MHz, DMSO-*d*_6_) δ 8.99 (s, 1H), 8.64 (d, *J* = 2.2 Hz, 1H), 8.56 (t, *J* = 5.6 Hz, 1H), 8.48 (s, 1H), 8.43 (s, 1H), 8.18–8.11 (m, 1H), 8.05 (dd, *J* = 8.6, 5.6 Hz, 2H), 7.87–7.65 (m, 3H), 7.29 (t, *J* = 8.8 Hz, 2H), 3.46 (t, *J* = 6.1 Hz, 2H), 1.22 (td, *J* = 11.8, 4.7 Hz, 1H), 0.57–0.45 (m, 2H), 0.32 (q, *J* = 5.0 Hz, 2H). ^13^C NMR (150 MHz, DMSO) δ 162.8, 161.1, 159.5, 155.4, 148.7, 144.2, 133.7, 130.7, 130.3, 128.3, 127.6 (2C), 125.2, 124.6, 124.3, 120.3, 116.7, 115.7 (2C), 115.2, 109.5, 45.2, 10.6, 3.6 (2C). ^19^F NMR (565 MHz, DMSO) δ −114.24. HRMS (ESI): calcd for C_25_H_21_N_5_F [M + H]^+^
*m*/*z* 410.1776, found 410.1773.

##### Compound 6-(2-phenylimidazo[1,2-a]pyridin-6-yl)-N-(1H-pyrazol-3-yl)quinazolin-4-amine (**13f**)

Pink solid, 54.6% yield, m.p. 289.1–290.1 °C. ^1^H NMR (600 MHz, DMSO-*d*_6_) δ 12.61 (s, 1H), 10.86 (s, 1H), 9.09 (s, 1H), 9.06 (s, 1H), 8.66 (s, 1H), 8.46 (s, 1H), 8.26 (d, *J* = 8.7 Hz, 1H), 8.03 (d, *J* = 7.6 Hz, 2H), 7.93 (d, *J* = 9.4 Hz, 1H), 7.88 (d, *J* = 8.6 Hz, 1H),7.83–7.68 (m, 2H), 7.48 (t, *J* = 7.6 Hz, 2H), 7.36 (t, *J* = 7.3 Hz, 1H), 6.91 (s, 1H). ^13^C NMR (150 MHz, DMSO) δ 154.4, 149.4, 148.1, 145.0, 144.2, 134.4, 133.6, 131.2, 129.0, 128.8 (2C), 128.0, 127.8, 125.7 (2C), 125.2, 124.6, 124.3, 120.6, 116.6, 116.6, 115.2, 109.7, 98.4. HRMS (ESI): calcd for C_24_H_18_N_7_ [M + H]^+^
*m*/*z* 404.1618, found 404.1613; C_24_H_17_N_7_Na [M + Na]^+^
*m*/*z* 426.1438, found 426.1426.

##### Compound 6-(2-cyclopropylimidazo[1,2-a]pyridin-6-yl)-N-(cyclopropylmethyl)quinazolin-4-amine (**13g**)

Yellow solid, 60.6% yield, m.p. 118.6–120.5 °C. ^1^H NMR (600 MHz, DMSO-*d*_6_) δ 8.89 (s, 1H), 8.60 (d, *J* = 2.1 Hz, 1H), 8.55 (t, *J* = 5.6 Hz, 1H), 8.46 (s, 1H), 8.09 (dd, *J* = 8.6, 2.0 Hz, 1H), 7.79 (s, 1H), 7.75 (d, *J* = 8.7 Hz, 1H), 7.68 (dd, *J* = 9.4, 1.9 Hz, 1H), 7.57 (d, *J* = 9.3 Hz, 1H), 3.45 (t, *J* = 6.2 Hz, 2H), 2.05 (ddd, *J* = 13.2, 8.3, 4.9 Hz, 1H), 1.25–1.18 (m, 1H), 0.92 (dt, *J* = 8.2, 2.9 Hz, 2H), 0.88–0.84 (m, 2H), 0.52–0.45 (m, 2H), 0.34–0.27 (m, 2H). ^13^C NMR (150 MHz, DMSO) δ 159.4, 155.2, 149.3, 148.5, 143.4, 134.0, 130.7, 128.2, 124.0, 123.8, 123.7, 120.1, 115.9, 115.2, 109.2, 45.2, 10.6, 9.5, 8.3 (2C), 3.6 (2C). HRMS (ESI): calcd for C_22_H_22_N_5_ [M + H]^+^
*m*/*z* 356.1870, found 356.1863; C_22_H_21_N_5_Na [M + Na]^+^
*m*/*z* 378.1690, found 378.1682.

##### Compound 6-(2-cyclopropylimidazo[1,2-a]pyridin-6-yl)-N-ethyl-N-phenylquinazolin-4-amine (**13h**)

Yellow solid, 58.3% yield, m.p. 166.8–168.7 °C. ^1^H NMR (600 MHz, DMSO-*d*_6_) δ 8.72 (s, 1H), 8.15 (s, 1H), 7.94 (dd, *J* = 8.7, 2.1 Hz, 1H), 7.80 (d, *J* = 8.6 Hz, 1H), 7.67 (s, 1H), 7.55 (t, *J* =7.3 Hz, 2H), 7.53–7.50 (m, 1H), 7.34 (d, *J* = 7.5 Hz, 3H), 7.04 (d, *J* = 2.1 Hz, 1H), 6.76 (dd, *J* = 9.3, 1.9 Hz, 1H), 4.18 (q, *J* = 7.0 Hz, 2H), 1.99–2.05 (m, 1H), 1.23 (t, *J* = 7.0 Hz, 3H), 0.90 (dt, *J* = 8.2, 2.9 Hz, 2H), 0.85–0.81 (m, 2H). ^13^C NMR (150 MHz, DMSO) δ 160.0, 154.2, 150.8, 149.4, 145.9, 143.2, 132.6, 130.4 (2C), 130.4, 128.8, 127.2 (2C), 127.1, 123.5, 123.4, 123.3, 122.9, 115.8, 115.8, 109.1, 48.1, 11.7, 9.4, 8.2 (2C). HRMS (ESI): calcd for C_26_H_24_N_5_ [M + H]^+^
*m*/*z* 406.2026, found 406.2019; C_26_H_23_N_5_Na [M + Na]^+^
*m*/*z* 428.1846, found 428.1840.

##### N-ethyl-N-phenyl-6-(2-phenylimidazo[1,2-a]pyridin-6-yl)quinazolin-4-amine (**13i**)

Pink solid, 67.4% yield, m.p. 203.0–205.1 °C. ^1^H NMR (600 MHz, DMSO-*d*_6_) δ 8.74 (s, 1H), 8.32 (s, 1H), 8.26 (s, 1H), 8.03–7.97 (m, 3H), 7.83 (d, *J* = 8.6 Hz, 1H), 7.61–7.57 (m, 3H), 7.54 (d, *J* = 9.3 Hz, 1H), 7.45 (t, *J* = 7.6 Hz, 2H), 7.38 (dd, *J* = 6.8, 2.8 Hz, 2H), 7.34 (t, *J* = 7.3 Hz, 1H), 7.07 (d, *J* = 2.0 Hz, 1H), 6.87 (dd, *J* = 9.4, 1.9 Hz, 1H), 4.21 (q, *J* = 7.0 Hz, 2H), 1.25 (t, *J* = 7.0 Hz, 3H). ^13^C NMR (150 MHz, DMSO) δ 160.0, 154.1, 150.5, 145.8, 145.2, 143.9, 133.6, 132.4, 130.5 (2C), 128.7 (2C), 128.6, 127.9, 127.4, 127.2 (2C), 125.7 (2C), 124.3, 124.1, 123.9, 123.7, 116.6, 115.7, 109.5 (2C), 48.2, 11.7. HRMS (ESI): calcd for C_29_H_24_N_5_ [M + H]^+^
*m*/*z* 442.2026, found 442.2021.

##### Compound 6-(2-phenylimidazo[1,2-a]pyridin-6-yl)-N-(pyridin-2-ylmethyl)quinazolin-4-amine (**13j**)

Pink solid, 70.2% yield, m.p. 141.9–143.3 °C. ^1^H NMR (600 MHz, DMSO-*d*_6_) δ 9.13 (t, *J* = 5.9 Hz, 1H), 9.02 (s, 1H), 8.75 (s, 1H), 8.54 (d, *J* = 4.3 Hz, 1H), 8.46 (s, 1H), 8.44 (s, 1H), 8.19 (dd, *J* = 8.7, 2.0 Hz, 1H), 8.01 (d, *J* = 7.5 Hz, 2H), 7.84–7.78 (m, 2H), 7.78–7.70 (m, 2H), 7.46 (t, *J* = 7.6 Hz, 2H), 7.39 (d, *J* = 7.9 Hz, 1H), 7.34 (t, *J* = 7.4 Hz, 1H), 7.29–7.25 (m, 1H), 4.93 (d, *J* = 5.8 Hz, 2H). ^13^C NMR (150 MHz, DMSO) δ 159.6, 158.6, 155.2, 149.0, 148.6, 145.2, 144.2, 136.8, 133.9, 133.7, 130.9, 128.8 (2C), 128.3, 127.9, 125.7 (2C), 125.0, 124.4, 124.4, 122.2, 121.2, 120.3, 116.8, 115.2, 109.7, 45.7. HRMS (ESI): calcd for C_27_H_21_N_6_ [M + H]^+^
*m*/*z* 429.1822, found 429.1817.

##### Compound 6-(2-phenylimidazo[1,2-a]pyridin-6-yl)-N-((tetrahydro-2H-pyran-4-yl)methyl)quinazolin-4-amine (**13k**)

Pink solid, 69.8% yield, m.p. 127.0–129.3 °C. ^1^H NMR (600 MHz, DMSO-*d*_6_) δ 8.98 (s, 1H), 8.64 (s, 1H), 8.51 (t, *J* = 5.8 Hz, 1H), 8.49 (s, 1H), 8.44 (s, 1H), 8.14 (dd, *J* = 8.6, 2.0 Hz, 1H), 8.01 (d, *J* = 7.5 Hz, 2H), 7.80–7.74 (m, 3H), 7.46 (t, *J* = 7.6 Hz, 2H), 7.34 (t, *J* = 7.3 Hz, 1H), 3.86 (d, *J* = 11.2 Hz, 2H), 3.49 (t, *J* = 6.4 Hz, 2H), 3.28 (t, *J* = 11.0 Hz, 2H), 2.06–1.97 (m, 1H), 1.67 (d, *J* = 10.4 Hz, 2H), 1.32–1.24 (m, 2H). ^13^C NMR (150 MHz, DMSO) δ 159.7, 155.3, 148.5, 145.2, 144.2, 133.8, 133.8, 130.8, 128.8 (2C), 128.2, 127.9, 125.7 (2C), 125.1, 124.5, 124.3, 120.3, 116.8, 115.2, 109.7, 66.8 (2C), 46.3, 34.2, 30.7 (2C). HRMS (ESI): calcd for C_27_H_26_ON_5_ [M + H]^+^
*m*/*z* 436.2132, found 436.2126.

### 4.2. Biological (Pharmacological) Research

#### 4.2.1. Cell Culture

Human cell lines HCC827, A549, SH-SY5Y, HEL, MCF-7 and MRC-5 obtained from the Chinese Academy of Sciences Cell Bank (Shanghai, China) were treated with 10% foetal bovine serum (FBS, Biological Industries, Cromwell, CT, USA) and 1% antibiotics-antimycotics (100 units/mL penicillin G sodium, 100 μg/mL streptomycin, and 250 ng/mL amphotericin B) added to RPMI-1640 (HCC827, SH-SY5Y, HEL) or DMEM (A549, MCF-7, MRC-5) in culture. Cells were grown at 37 °C in an incubator containing water and 5% CO_2_.

#### 4.2.2. Antiproliferative Activity Assay

Cells were seeded in 96-well plates at 3000–5000 cells/well and treated with different concentrations of compounds for 72 h. After treatment, 20 μL MTT (Sigma-Aldrich, St. Louis, MO, USA) was added to each well, and incubation was continued in the incubator for 4 h. Purple formazan crystals were formed, the medium was discarded, 150 μL DMSO was added to dissolve the formazan, and the absorbance at 490 nm was measured by a multi-well spectrophotometer (Thermo Scientific, VARIOSKAN LUX, Waltham, MA, USA) to measure absorbance at 490 nm and to measure viability. IC_50_ values were calculated based on the inhibition rate using GraphPad Prism software.

#### 4.2.3. Molecular Modelling

Molecular docking simulations were performed using Molecular Operating Environment (MOE, Version 2020) [52]. PI3Kα (PDB code: 4ZOP) is selected for docking studies. Protein optimisation was performed by quickprep of the MOE. Docking sites were defined by the Site Finder program and Accelrys Discovery Studio Visualizer 4.5 was used for graphical display.

#### 4.2.4. Kinase Assay

The inhibitory activity of compound **13k** against PI3Kα was determined using the ADP-GloTM Max Assay, with HS-173 as a positive control, according to the kit instructions. Chemiluminescence values were measured by multi-well spectrophotometer (Thermo Scientific, VARIOSKAN LUX, USA).

#### 4.2.5. Cell Cycle Assays

HCC827 cells were incubated in 6-well plates and treated with specific concentrations of **13k** for 48 h. Cells were collected and washed with PBS buffered solution, fixed overnight at −20 °C with pre-cooled 70% ethanol, supernatant discarded, washed with PBS buffered solution, stained by a mixture of propidium iodide (PI) and RNase, incubated for 30 min at room temperature protected from light and then detected using flow cytometry.

#### 4.2.6. Hochest 33342 Staining Assay

A portion of HCC827 cells were taken and inoculated overnight in 6-well plates and treated with different concentrations of compound **13k** for 48 h. Subsequent steps were carried out according to the instructions of the Hochest 33342 staining kit (Beyotime, Shanghai, China). Final pictures were taken with a microscope (DMi8, Leica, Wetzlar, Germany).

#### 4.2.7. Apoptosis Assay

Apoptosis was detected by flow cytometry after staining with Annexin V-FITC and propidium iodide (PI) according to the manufacturer’s protocol (BD Biosciences). HCC827 cells were inoculated overnight in 6-well plates, treated with specific concentrations of compound **13k** for 48 h. Cells were collected and incubated with 5 μL of membrane linked protein V-FITC and 5 μL of PI for 15–20 min protected from light, followed by flow cytometry analysis.

#### 4.2.8. Western Blot Assay

Cells were treated with different concentrations of compound **13k** and then subjected to immunoblot analysis as described in a previous study. Blots were imaged by a ChemiDoc^TM^ MP imaging system (Bio-Rad, Hercules, CA, USA). All bands were analysed using Image J software. Antibodies were purchased from Cell Signaling Technology (CST, Danvers, MA, USA).

#### 4.2.9. 3D Spheroid Cell Inhibition Assay

To culture HCC827 cancer cells into three-dimensional spheroids, we used PerkinElmer’s CellCarrier Spheroid ULA 96-well microtiter plates (PerkinElmer, Waltham, MA, USA). In all experiments, cells were seeded at 40,000 cells per well. After spheroid formation, the spheroids were treated with **13k** at the indicated concentrations every 3 days. When significant changes in tumour spheroids were observed, photographs were taken using a ZEISS LSM 900 Airyscan 2 confocal laser scanning microscopy (ZEISS, Jena, Germany).

#### 4.2.10. Statistical Analysis

All experimental data were replicated three times, and experimental results are expressed as mean ± standard deviation (SD). Statistical analyses were manipulated and plotted using Photoshop, ImageJ, Graph Pad, etc., and tests were performed to assess statistically significant differences (* *p* < 0.05, ** *p* < 0.01, *** *p* < 0.001 or n.s. (not significant)).

## Data Availability

All data are included within the article and Appendix A.

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
