# Peer review of "Design, Synthesis and Biological Evaluation of 6-(Imidazo[1,2-a]pyridin-6-yl)quinazoline Derivatives as Anticancer Agents via PI3Kα Inhibition"

_ijms, 2023, doi:10.3390/ijms24076851_

Round 1

Reviewer 1 Report

The authors well described the biological activity of newly synthesized PI3K inhibitors that were designed on the basis of their previous studies. There are minor points that authors can modify or address to improve the manuscript.

1. Abstract (line18-19): minor English correction : interesting for cancer therapy-> interests for the treatment of cancer. 

2.  As the authors used HCC827 cell lines for biological evaluation, please include the rationale for using PI3K inhibitors in NSCLC cells in the discussion. 

3.In Figure 1, the IC50 against PI3Ka of HS-173 is 0.8 nM. However, the IC50 against PI3Ka of HS-173 in Table 3 is 3.72 nM. Please confirm and correct the data accordingly.

4. Compound 13k is sometimes written in bold style and sometimes in plain text. Please make the style consistent.

Author Response

We would like to appreciated the reviewers for considering our manuscript carefully and for their insightful and profound suggestions.

After fundamentally revising of present study and article wording modification, we resubmitted the manuscript. And the detailed, point-by-point response to reviewers’ and editor’s comments were as follows.

Reviewer:

1. Abstract (line18-19): minor English correction: interesting for cancer therapy-> interests for the treatment of cancer. 

Response: Thank you very much for your careful review and this comment. This part had been modified as advised. See the section of “line 16”.

2. As the authors used HCC827 cell lines for biological evaluation, please include the rationale for using PI3K inhibitors in NSCLC cells in the discussion. 

Response: Thank you very much for your careful review and this comment. This part had been modified as advised. See the section of “line 123-125 and line 142-147”.

3.In Figure 1, the IC50 against PI3Ka of HS-173 is 0.8 nM. However, the IC50 against PI3Ka of HS-173 in Table 3 is 3.72 nM. Please confirm and correct the data accordingly.

Response: Thank you very much for your careful review and this comment. This section of data has been corrected.

4. Compound 13k is sometimes written in bold style and sometimes in plain text. Please make the style consistent.

Response: Thank you very much for your careful review and this comment. This part had been modified as advised.

Responses to reviewers are uploaded as a word file.

Reviewer 2 Report

Dear Authors,

Before acceptance, this paper needs several corrections.

My comments are listed below.

L. 14-18 - The sentence: ‘Phosphatidylinositol 3 kinase (PI3K) is an important intracellular signal transduction molecule involved …’ should be in the Introduction.

L. 19-20 - Please correct or clarify the following sentence: ‘In this study, a series of novel 6-(imidazo[1,2-a]pyridin-6-yl)quinazoline derivatives were designed, synthesized and identified as PI3Kα inhibitor through biological investigation.’

L. 38 - Please explain the gene acronym ‘PTEN’

L. 56-58 - Please correct or clarify the following sentences ‘And a series of 6-(imidazo[1,2-a]pyridin-6-yl)quinazoline derivatives were designed, synthesised and evaluated as PI3Kα inhibitors. According to the SAR analysis, 4-aminoquinazoline derivative moiety is the main critical pharmacophore of 6b for its PI3Kα inhibitory activity.’

L. 62 - Please explain the phrase ‘thickened heterocyclic compounds’

L. 65-68 - ‘Based on the data of biological evaluation, compound 13k was found to show the best antiproliferative activity against all the test cancer cells, and induced cell cycle arrest in G2/M phase and cell apoptosis by inhibition of PI3Kα.’ - This is the result of this research. Why is it reported in the Introduction?

L. 87 - ‘synthesised various lipids’ - it is not clear.

L. 99 - ‘2.2.1. Antiproliferative assays in vitro’ - The description of the biological part needs improvement. Authors should compare the two series of compounds: 10a-n (containing R3 = COOC2H5) with 10o-u (containing R3 = COOCH3). Moreover, in comparison of the 10a-m series and 10o-10u series (e.g. 10h and 10q), the Authors should choose compounds with the same R2 substituent. In the same way, more compounds should be compared in the series 13a-k. Why the Authors chose only compounds 13j and 13k? In my opinion, it does not allow you to draw any general conclusions about the biological activity of tested compounds.

L. 291 - I suggest that the authors should give the general synthetic procedure for compounds 7a-k.

L. 333 - In the same manner, the general synthesis procedure for compounds 10a-u should be given.

In the description of the 1H-NMR spectrum please verify some coupling constants (e.g. L. 404 and 415 ‘J=23.8 Hz’ and ‘J=30.3 Hz’. In the description of the 13C-NMR spectra add some overlapping signals of carbon atoms. It would be useful for the reader to give the relevant general description of the NMR spectra for the series of compounds 10 and 13.

References - The reviewer believes that the DOI must be completed.

Author Response

We would like to appreciated the reviewers for considering our manuscript carefully and for their insightful and profound suggestions.

After fundamentally revising of present study and article wording modification, we resubmitted the manuscript. And the detailed, point-by-point response to reviewers’ and editor’s comments were as follows.

Reviewer:

1.L. 14-18 - The sentence: ‘Phosphatidylinositol 3 kinase (PI3K) is an important intracellular signal transduction molecule involved …’ should be in the Introduction.

Response: Thank you very much for your careful review and this comment. This part had been modified as advised. See the section of “line 14-15”.

2. L. 19-20 - Please correct or clarify the following sentence: ‘In this study, a series of novel 6-(imidazo[1,2-a]pyridin-6-yl)quinazoline derivatives were designed, synthesized and identified as PI3Kα inhibitor through biological investigation.’

Response: Thank you very much for your careful review and this comment. This part had been modified as advised. See the section of “line 16-18”.

3. L. 38 - Please explain the gene acronym ‘PTEN’

Response: Thank you very much for your careful review and this comment. This part had been modified as advised. See the section of “line 35”.

4. L. 56-58 - Please correct or clarify the following sentences ‘And a series of 6-(imidazo[1,2-a]pyridin-6-yl)quinazoline derivatives were designed, synthesised and evaluated as PI3Kα inhibitors. According to the SAR analysis, 4-aminoquinazoline derivative moiety is the main critical pharmacophore of 6b for its PI3Kα inhibitory activity.’

Response: Thank you very much for your careful review and this comment. This part had been modified as advised. See the section of “line 53-55 and line 59-61”.

5. L. 62 - Please explain the phrase ‘thickened heterocyclic compounds’

Response: Thank you very much for your careful review and this comment. I'm sorry, it was a mistake, it is now corrected to ‘nitrogen-containing fused heterocyclics’.

6. L. 65-68 - ‘Based on the data of biological evaluation, compound 13k was found to show the best antiproliferative activity against all the test cancer cells, and induced cell cycle arrest in G2/M phase and cell apoptosis by inhibition of PI3Kα.’ - This is the result of this research. Why is it reported in the Introduction?

Response: Thank you very much for your careful review and this comment. This part had been removed.

7. L. 87 - ‘synthesised various lipids’ - it is not clear.

Response: Thank you very much for your careful review and this comment. This part had been modified. See the section of “line 80”.

8. L. 99 - ‘2.2.1. Antiproliferative assays in vitro’ - The description of the biological part needs improvement. Authors should compare the two series of compounds: 10a-n(containing R3 = COOC2H5) with 10o-u(containing R3 = COOCH3). Moreover, in comparison of the 10a-m series and 10o-10u series (e.g. 10h and 10q), the Authors should choose compounds with the same R2 substituent. In the same way, more compounds should be compared in the series 13a-k. Why the Authors chose only compounds 13j and 13k? In my opinion, it does not allow you to draw any general conclusions about the biological activity of tested compounds.

Response: Thank you very much for your careful review and this comment. This part had been modified as advised. See the section of “line 110-118”.

9. L. 291 - I suggest that the authors should give the general synthetic procedure for compounds 7a-k.

Response: Thank you very much for your careful review and this comment. This part had been modified as advised. Due to the length of the article, this section of data has been added to the supporting information.

10. L. 333 - In the same manner, the general synthesis procedure for compounds 10a-ushould be given.

Response: Thank you very much for your careful review and this comment. This part had been modified as advised. Due to the length of the article, this section of data has been added to the supporting information.

11. In the description of the 1H-NMR spectrum please verify some coupling constants (e.g. L. 404 and 415 ‘J=23.8 Hz’ and ‘J=30.3 Hz’. In the description of the13C-NMR spectra add some overlapping signals of carbon atoms. It would be useful for the reader to give the relevant general description of the NMR spectra for the series of compounds 10and 13.

Response: Thank you very much for your careful review and this comment. This part had been modified as advised.

12. References- The reviewer believes that the DOI must be completed

Response: Thank you very much for your careful review and this comment. This part had been modified as advised.

Responses to reviewers are uploaded as a word file.

Reviewer 3 Report

The authors in the current manuscript evaluated a better version of the PI3K inhibitor, which is a continuum of their previous article. I am not sure how this article is adding to the existing literature and knowledge and see this as an extension. Here are a few points that can improve the article and make its own place apart from the previously published articles,

1.       The authors need to test the compound along with positive control in mice (tumor formation assay, assessing the effect of their lead molecule on mice health and living)

2.       The authors should test other kinase pathways (like MAPK and p38) along with the PI3K-AKT-mTOR (Fig 2).

3.       Which phosphorylation the authors test for AKT; they should test both the phosphorylation.

4.       The authors should also test the time-dependent activity of 13K.

5.       The authors should clarify how they think BRD4 can be involved in regulating the cell cycle with respect to 13k or lese can remove the data.

6.       Does 13K induce DNA instability and damage?

7.       The author's shroud assesses the BLc2/Bax ratio.

8.       The quality of the images needs to be improved, does 13K somehow affect the intensity of Hoechst staining (Fig 5A)?

9.       The authors should discuss why they think their study is important, what’s the key finding and why its novel.

10.   English needs to be improved. 

Author Response

We would like to appreciated the reviewers for considering our manuscript carefully and for their insightful and profound suggestions.

After fundamentally revising of present study and article wording modification, we resubmitted the manuscript. And the detailed, point-by-point response to reviewers’ and editor’s comments were as follows.

Reviewer:

1. The authors need to test the compound along with positive control in mice (tumor formation assay, assessing the effect of their lead molecule on mice health and living)

Response: Thank you very much for your careful review and this valuable suggestion. As the reviewer suggested, it would be better if we tested the effects of our lead compound as well as the positive control in vivo. Unfortunately, we can’t complete this work so far because it will take one and a half months to test the efficacy in vivo. Moreover, we need corresponding synthetic raw materials for the synthesis of the active compounds for in vivo experiments, but due to change in purchase regulations, we cannot purchase all the synthetic raw materials and reagents at present. However, we will conduct relevant in vivo studies in subsequent studies.

2. The authors should test other kinase pathways (like MAPK and p38) along with the PI3K-AKT-mTOR (Fig 2).

Response: Thank you very much for your careful review and this comment. This part had been modified as advised. See the section of “line 152-158 and line 168-173”.

3. Which phosphorylation the authors test for AKT; they should test both the phosphorylation.

Response: Thanks a lot for your valuable comments. As the reviewer said, we can more fully elucidate the full activation of Akt if we examined the phosphorylation at both Thr308 and Ser473. However, we only tested the phosphorylation at Ser 473 of Akt in this study because we contacted several companies to purchase p-Akt (Thr308) antibodies, but none were in stock. In addition, there are studies indicated phosphorylation of Akt (Ser473) is already an excellent predictor of Akt activation.

4. The authors should also test the time-dependent activity of 13k.

Response: Thank you very much for your careful review and this comment. This part had been modified as advised. See the section of “line 123-125 and line 131-134”.

5. The authors should clarify how they think BRD4 can be involved in regulating the cell cycle with respect to 13k or lese can remove the data.

Response: Thank you very much for your careful review and this comment. This part data has been removed.

6. Does 13K induce DNA instability and damage?

Response: PARP has been found to play important roles in the DNA damage response. In our study, 13k could increase the levels of cleaved-PARP. So, we believed 13k induced DNA instability and damage.

7. The author's shroud assesses the BLc2/Bax ratio.

Response: Thank you very much for your careful review and this comment. This part had been modified as advised. See the section of “line 218-227”.

8. The quality of the images needs to be improved, does 13K somehow affect the intensity of Hoechst staining (Fig 5A)?

Response: Thank you very much for your comments. Figure 5 has been corrected to Figure7, and we have improved the image quality of Figure 7A. See the section of “line 221”. As shown in Figure 7, 13k treatment led to cell apoptosis along with cell morphological alterations including cell shrinkage and nuclear fragmentation, which resulted in an enhanced Hoechst absorption and an intensity of Hoechst staining. 

9. The authors should discuss why they think their study is important, what’s the key finding and why its novel.

Response: Thank you very much for your careful review and this comment. We have made corresponding modifications according to the reviewer's comments. See the section of “line237-239”.

10. English needs to be improved. 

Response: We have made corresponding modifications according to the reviewer's comments and the modifications have been highlighted in yellow in the manuscript.

Responses to reviewers are uploaded as a word file.

Round 2

Reviewer 2 Report

Dear Authors,

The resubmitted manuscript deserves publication in the International Journal of Molecular Sciences.

Author Response

Ms. Ref. No.: ijms-2253333

Title: Design, synthesis and biological evaluation of 6-(imidazo[1,2-a]pyridin-6-yl)quinazoline derivatives as anticancer agents via PI3Kα inhibition

Dear reviewer,

We gratefully thank the reviewers for their time spend making their constructive remarks and useful suggestions, which has significantly raised the quality of the manuscript and has enable us to improve the manuscript.

Reviewer 3 Report

The authors have done some of the experiments, however, the animal experiments and the analog testing are critical for the manuscript to have a significant impact and acceptance in a wider community. 

Author Response

Ms. Ref. No.: ijms-2253333

Title: Design, synthesis and biological evaluation of 6-(imidazo[1,2-a]pyridin-6-yl)quinazoline derivatives as anticancer agents via PI3Kα inhibition

Dear reviewer,

We would like to appreciate the reviewers for considering our manuscript carefully and for their insightful and profound suggestions.

After fundamentally revising of present study and article wording modification, we resubmitted the manuscript. And the detailed, response to reviewer comments were as follows.

Reviewer:

The authors have done some of the experiments, however, the animal experiments and the analog testing are critical for the manuscript to have a significant impact and acceptance in a wider community.

Response: Thank you very much for your careful review and this comment. We refer to the following literature and found that the 3D tumor spheroid models exhibit heterogeneity and structural complexity reflecting in vivo tumors, resulting in a more accurate representation of the tumor physiology occurring in oncologic disease. Hence, it is widely used to investigate the antitumor efficacy of the active compounds in early studies of drug development. Therefore, we construct a 3D tumor sphere formation model to examine the antitumor efficacy of our target compound. When treated with increased concentration of compound 13k, indicating that 13k could effectively inhibit the tumor sphere formation and has potential for further preclinical studies. This section has been modified in the “Results and Discussion” section.

  1. Han, K.; Pierce, S. E.; Li, A.; Spees, K.; Anderson, G. R.; Seoane, J. A.; Lo, Y. H.; Dubreuil, M.; Olivas, M.; Kamber, R. A.; Wainberg, M.; Kostyrko, K.; Kelly, M. R.; Yousefi, M.; Simpkins, S. W.; Yao, D.; Lee, K.; Kuo, C. J.; Jackson, P. K.; Sweet-Cordero, A.; Kundaje, A.; Gentles, A. J.; Curtis, C.; Winslow, M. M.; Bassik, M. C., CRISPR screens in cancer spheroids identify 3D growth-specific vulnerabilities. Nature 2020, 580, 136-141, https://doi.org/10.1038/s41586-020-2099-x.
  2. Koledova, Z., 3D Cell Culture: An Introduction. Methods. Mol. Biol. 2017, 1612, 1-11, https://doi.org/10.1007/978-1-4939-7021-6_1.
  3. Langhans, S. A., Using 3D in vitro cell culture models in anti-cancer drug discovery. Expert. Opin. Drug. Discov. 2021, 16, 841-850, https://doi.org/10.1080/17460441.2021.1912731.
  4. Xiao, Z.; Osipyan, A.; Song, S.; Chen, D.; Schut, R. A.; van Merkerk, R.; van der Wouden, P. E.; Cool, R. H.; Quax, W. J.; Melgert, B. N.; Poelarends, G. J.; Dekker, F. J., Thieno[2,3-d]pyrimidine-2,4(1H,3H)-dione Derivative Inhibits d-Dopachrome Tautomerase Activity and Suppresses the Proliferation of Non-Small Cell Lung Cancer Cells. J. Med. Chem. 2022, 65, 2059-2077, https://doi.org/10.1021/acs.jmedchem.1c01598.
